# Scaling and similarity of a stream-power incision and linear diffusion landscape evolution model

Nikos Theodoratos<sup>1</sup>, Hansjörg Seybold<sup>1</sup>, and James W. Kirchner<sup>1,2</sup>

<sup>1</sup>Dept. of Environmental Systems Science, ETH Zurich, Zurich, 8092, Switzerland 5

<sup>2</sup>Swiss Federal Research Institute WSL, Birmensdorf, 8903, Switzerland

Correspondence to: Nikos Theodoratos (theodoratos@usys.ethz.ch)

Abstract. Scaling and similarity of fluvial landscapes can reveal fundamental aspects of the physics driving their evolution. Here we perform dimensional analysis on the governing equation of a widely used landscape evolution model (LEM) that combines stream-power incision and linear diffusion laws. Our analysis assumes that length and height are conceptually

- 10 distinct dimensions, and uses characteristic scales that depend only on the model parameters (incision coefficient, diffusion coefficient, and uplift rate) rather than on the size of the domain or of landscape features. We use previously defined characteristic length, height, and time scales, but, for the first time, we combine all three in a single analysis. Using these characteristic scales, we non-dimensionalize the LEM such that it includes only dimensionless variables and no parameters. This significantly simplifies the LEM by removing all parameter-related degrees of freedom. The only remaining degrees of
- freedom are in the boundary and initial conditions. Thus, for any given set of dimensionless boundary and initial conditions, 15 all simulations, regardless of parameters, are just re-scaled copies of each other, both in steady state and throughout their evolution. Therefore, the entire model parameter space can be explored by temporally and spatially re-scaling a single simulation. This is orders of magnitude faster than performing multiple simulations to span multi-dimensional parameter spaces.

The characteristic length, height, and time scales are geomorphologically interpretable; they define relationships between topography and the relative strengths of landscape-forming processes. The characteristic height scale specifies how drainage areas and slopes must be related to curvatures for a landscape to be in steady state, and leads to methods for defining valleys, estimating model parameters, and testing whether real topography follows the LEM. The characteristic length scale is roughly equal to the scale of the transition from diffusion-dominated to advection-dominated propagation of topographic

  - perturbations (e.g., knickpoints). We introduce a modified definition of the landscape Péclet number, which quantifies the relative influence of advective versus diffusive propagation of perturbations. Our Péclet number definition can account for the scaling of basin length with basin area, which depends on topographic convergence versus divergence.

# 1 Introduction

Hillslopes and river valleys are organized in striking patterns that appear to be repeated across landscapes and scales. Furthermore, within each landscape the transition from hillslopes to valleys seems to occur at a characteristic scale. These two properties have captivated scientists from the early days of geomorphology (e.g., Gilbert, 1877; Davis, 1892). Both

properties are thought to be related to the scaling of processes that shape fluvial landscapes (e.g., Perron et al., 2008, 2012; Paola et al., 2009).

Scaling problems are often studied with the aid of dimensional analysis (e.g., Sonin, 2001; Bear and Cheng, 2010) which

- stems from Fourier's principle that all terms of physically meaningful equations should have consistent dimensions (Huntley, 1967). Dimensional analyses of landscape evolution models (LEMs) have been used to describe how the relative strengths of landscape-forming processes control properties of ridge and valley topography, such as drainage density (Willgoose et al., 1991), shapes of basins (Howard, 1994), fluvial relief of mountains (Whipple and Tucker, 1999), topographic roughness (Simpson and Schlunegger, 2003), valley spacing (Perron et al., 2008, 2009), and drainage areas of first- and second-order
- valleys (Perron et al., 2012). The aforementioned studies used LEMs that, while differing in details, all assumed that fluvial landscapes are shaped by a combination of advective and diffusive erosion, with the former dominating valleys and the latter dominating hillslopes.

Here we present a dimensional analysis of the governing equation of a simple, widely used LEM (see Eq. 1 below). Our

- work is based on two key premises. First, we define characteristic scales from the model parameters, rather than from extrinsic properties of the simulated landscape, such as the domain size or relief. Characteristic scales defined in this way are intrinsic to each landscape and its parameters (and thus to its underlying properties and to the strengths of the processes that shape it), and are independent of the initial and boundary conditions of any simulation. Second, in our approach we distinguish between the dimensions of horizontal length and vertical height. These two premises are not new (e.g., Willgoose
- et al., 1991; Whipple and Tucker, 1999; Perron et al., 2008; Robl et al., 2017), but here, for the first time, we apply them jointly to define and interpret all of the characteristic scales in this LEM. In so doing, we obtain three characteristic scales a characteristic length, height, and time that significantly simplify the model's governing equation. These characteristic scales are also geomorphologically interpretable, linking competition between processes to the scales of the features that emerge from these processes.

Our specific results directly apply only to the LEM that we dimensionally analyzed and to the landscapes that we can assume to be described by this LEM. However, our work illustrates an approach for the definition and interpretation of characteristic scales that could potentially be employed in dimensional analyses of other LEMs as well.

## 2 Theory

### 30 2.1 Landscape evolution model

We perform dimensional analysis of a simple, widely used model that describes the evolution of landscapes under the influence of stream-power incision, linear diffusion, and uplift, according to the governing equation

$$\frac{\partial z}{\partial t} = -K\sqrt{A}|\nabla z| + D\nabla^2 z + U, \qquad (1)$$

(e.g., Howard, 1994; Dietrich et al., 2003) where z(x, y) is the elevation of a point with horizontal coordinates (x, y), *t* is time and  $\partial z/\partial t$  is the rate of change of elevation, *K* is the coefficient of incision, *A* is the drainage area (per unit contour width) at the point (x, y), and  $|\nabla z|$  is the norm of the gradient of *z* (i.e., the topographic slope in the direction of steepest descent, to

which the gradient vector points by definition), D is the coefficient of diffusion and  $\nabla^2 z$  is the Laplacian of z (i.e., the topographic curvature, here assumed to be positive in concave-up areas, e.g., valleys, and negative in concave-down areas, e.g., hillslopes), and U is the uplift rate. The dimensions of the variables and parameters of Eq. (1) are discussed in the following subsection, which focuses on dimensional analysis.

The incision term  $-K\sqrt{A}|\nabla z|$  gives the rate of change of elevation due to detachment-limited sediment transport by flowing water, assumed to be proportional to stream power, i.e., the work performed by water per unit time, per unit streambed area (Dietrich et al., 2003). The incision term of Eq. (1) is a specific case of the more general incision term  $-KA^m|\nabla z|^n$ , which can take on different drainage area and slope exponents *m* and *n* in order to express more general incision behavior (e.g.,

- Dietrich et al., 2003). Here we assume m=0.5 and n=1, in keeping with the stream power law, and with the common assumptions that discharge scales linearly with drainage area and that channel width scales with the square root of drainage area, but in Appendix A we present results from dimensional analysis of the same LEM with generic exponents m and n (Eq. A1). We show that all the results and interpretations that we derived from the simplified LEM (Eq. 1), which we present in the main text, can be recovered from results of the generic LEM (Eq. A1) by setting m=0.5 and n=1. We do not present
- any results or interpretations that do not satisfy this condition. In other words, our results are not merely properties of a special case; we just describe them by simpler formulas whose presentation and interpretation is straightforward. Additionally, the incision term could include a threshold, below which no incision occurs (e.g., Dietrich et al., 2003; Perron et al., 2008). We investigate the scaling behavior of an LEM with an incision term with a non-zero incision threshold in other work, to be published separately.

The diffusion term  $D\nabla^2 z$  gives the rate of change of elevation due to diffusive sediment transport processes, such as soil creep due to bioturbation, freeze-thaw cycles, etc. The diffusion term is linear and assumes that the flux of sediment is proportional to the topographic slope (Fernandes and Dietrich, 1997). Depending on the sign of the curvature, this term can be either negative or positive, corresponding to erosional diffusion on ridges and hillslopes or depositional diffusion in valleys.

The term U gives the rate of increase of elevation relative to the boundary due to either a falling base level or tectonic uplift of the domain's interior. Throughout this paper, we refer to both effects as uplift since they are mathematically indistinguishable.

#### 30

If a landscape evolves to a condition in which, at every point, the three terms of the right-hand side of Eq. (1) cancel each other out, then  $\partial z/\partial t = 0$  everywhere; this condition is termed steady state or dynamic equilibrium. Numerical simulations are typically assumed to have converged to steady state when the rate of elevation change is everywhere smaller than an arbitrary, small threshold  $\varepsilon$ :

$$|\partial z/\partial t| \le \varepsilon \quad \forall (x, y) .$$
<sup>(2)</sup>

Equation (1) is presumed to describe soil-mantled landscapes with sufficiently cohesive soils and gentle slopes, where we can assume that incision is detachment-limited and diffusion is linear (e.g., Perron et al., 2008). Other types of landscapes are shaped by processes that cannot be described by Eq. (1). For example, diffusive processes on steep soil-mantled hillslopes are better described by a non-linear diffusion term (e.g., Roering et al., 1999, 2007).

Equation (1) assumes that all three processes act at all points of the landscape, without distinguishing between channels or hillslopes (e.g., Howard, 1994; Simpson and Schlunegger, 2003). All three terms are needed to model fluvial landscapes. Uplift fulfills the role of the source term, forcing the evolution of the landscape (e.g., Tucker and Hancock, 2010). Without

- it, the landscape would decay to a flat surface of zero elevation. The combination of the incision and diffusion terms is necessary for the emergence of ridges and valleys. Whereas the incision term amplifies topographic perturbations, setting in motion a positive feedback, the diffusion term dampens them, leading to a negative feedback (e.g., Smith and Bretherton, 1972; Perron et al., 2012). Both types of feedback are needed for the synthesis of surfaces with complex structures that resemble ridge and valley topography; therefore, the incision and diffusion terms of Eq. (1), or other terms with equivalent
- properties, represent the simplest combination of processes that can model landscapes characterized by ridge and valley topography (e.g., Smith and Bretherton, 1972; Howard, 1994). Because different points have different topographic properties (drainage area, slope, and curvature), the modeled incision and diffusion processes have different relative strengths across the landscape. Thus, even though distinct convergent (channel) and divergent (hillslope) landforms are not specified a priori, they can emerge from Eq. (1) (e.g., Howard, 1994), at scales that can be explored using dimensional analysis (e.g., Perron et al., 2008, 2009, 2012).

#### 2.2 Dimensional analysis

Our dimensional analysis of Eq. (1) begins by specifying the dimensions of its various variables. We will rescale Eq. (1) in the horizontal direction separately from the vertical direction so that we can study scaling of lengths and reliefs separately (e.g., Dietrich and Montgomery, 1998). Therefore, we must assume that the coordinates of points (x, y) have dimensions of

- length (L), while elevation *z* has an equivalent, but conceptually distinct, dimension of height (H). Huntley (1967) outlined a theoretical justification for distinct dimensions for quantities with identical units, and presented examples demonstrating the benefits of this approach. Distinct dimensions of length and height have been adopted by some previous studies of scaling of landscapes (e.g., Willgoose et al., 1991), but using different approaches than the one outlined below. The three fundamental dimensions in this model are length (L), height (H) and time (T), and all variables of Eq. (1) have dimensions that are
- powers, products, or ratios of L, H, and T. Specifically, the rate of elevation change  $\partial z/\partial t$  and, thus, also the incision, diffusion, and uplift terms of Eq. (1) have dimensions of H T<sup>-1</sup>. Given that the gradient  $|\nabla z|$  and the curvature  $\nabla^2 z$  have dimensions of H L<sup>-1</sup> and H L<sup>-2</sup>, respectively, the parameters *K*, *D*, and *U* must have dimensions of T<sup>-1</sup>, L<sup>2</sup> T<sup>-1</sup>, and H T<sup>-1</sup>, respectively.
- Because the dimensions of all variables of Eq. (1) can be expressed in terms of L, H, and T, we can non-dimensionalize Eq. (1) using a characteristic length  $l_c$ , a characteristic height  $h_c$ , a characteristic time  $t_c$ , and combinations thereof. For important reasons that are discussed throughout this study, we choose to define  $l_c$ ,  $h_c$ , and  $t_c$  as intrinsic scales, i.e., in terms of the model's parameters *K*, *D*, and *U*, not in terms of sizes of geomorphic features, such as basin length (e.g., Willgoose et al., 1991; Whipple and Tucker, 1999) or total relief (e.g., Whipple and Tucker, 1999; Perron et al., 2008), or in terms of the
- extent of the solution domain (e.g., Simpson and Schlunegger, 2003).

We define the characteristic length scale as

$$l_c \coloneqq \sqrt{D/K},\tag{3}$$

which has dimensions of length. Perron et al. (2008) showed that  $\sqrt{D/K}$  is related to the competition between incision and diffusion, and that it controls the scales of valley spacing (Perron et al., 2009) and valley-network branching (Perron et al., 2012). The quantity  $\sqrt{D/K}$  has also been previously used to non-dimensionalize LEMs (e.g., Duvall and Tucker, 2015).

We introduce a characteristic height scale

$$h_c \coloneqq U/_K, \tag{4}$$

which has dimensions of height, and a characteristic time scale

$$t_c \coloneqq \frac{1}{K}, \tag{5}$$

which has dimensions of time (these characteristic scales are equivalent to those presented by Robl et al. (2017) using a somewhat different formulation). The U/K ratio has been previously used in several different contexts. For instance,

- $(U/K)^{1/n}$ , where *n* is the slope exponent, is the steady-state value (e.g., Moglen and Bras, 1995; Sklar and Dietrich, 1998) of the steepness index (e.g., Whipple, 2001), which has been used to analyze river profiles and predict ridge migration dynamics (e.g., Wobus et al., 2006; Harkins et al., 2007; Whipple et al., 2017). It has also been previously noted that the U/K ratio scales relief (e.g., Sklar and Dietrich, 1998; Perron and Royden, 2013; Willett et al., 2014). To the best of our knowledge, however, previous dimensional analyses have not used U/K as a characteristic height scale to non-
- dimensionalize landscape evolution equations.

The characteristic length, height, and time scales  $l_c$ ,  $h_c$ , and  $t_c$ , as defined above (Eqs. 3–5), have not been previously combined to non-dimensionalize landscape evolution equations. Here, we adopt them as a group on purely dimensional grounds, because Eqs. (3), (4), and (5) define the only combinations of *D*, *K*, and *U* that yield dimensions of L, H, and T,

respectively. However, in Sect. 4 we also show that these characteristic scales have interesting geomorphological properties that further justify defining them according to Eqs. (3)–(5).

In Appendix A, we define analogous characteristic scales for an LEM with generic exponents m and n. In this more general case, each of the three characteristic scales depends on all three parameters K, D, and U.

Using the characteristic scales  $l_c$ ,  $h_c$ , and  $t_c$ , we can express length, height, and time in equivalent dimensionless forms:  $x = l_c x^*$ ,  $y = l_c y^*$ ,  $z = h_c z^*$ ,  $t = t_c t^*$ , (6) where the starred variables are dimensionless. Likewise, we can express each term of Eq. (1) as a corresponding dimensionless term multiplied by characteristic scales (or their products and ratios) which carry the corresponding dimensions.

## 30

Specifically, each differential (here using dz as an example) can be re-expressed as  $dz = h_c dz^*$ .

(7)

Consequently, we can express the rate of elevation change as

$$\frac{\partial z}{\partial t} = \frac{h_c}{t_c} \frac{\partial z^*}{\partial t^*} = U \frac{\partial z^*}{\partial t^*},\tag{8}$$

which suggests that we can view the uplift rate U as a characteristic rate of elevation change. Furthermore, we can express the gradient operator as

$$\nabla = \frac{\partial}{\partial x}\mathbf{i} + \frac{\partial}{\partial y}\mathbf{j} = \frac{1}{l_c} \left(\frac{\partial}{\partial x^*}\mathbf{i} + \frac{\partial}{\partial y^*}\mathbf{j}\right) = \frac{1}{l_c}\nabla^*, \qquad (9)$$

where **i** and **j** are the unit vectors in the direction of the *x* and *y* coordinates and  $\nabla^*$  is the gradient operator in dimensionless coordinates. Therefore, we can express topographic slope as

$$|\nabla z| = \frac{1}{l_c} |\nabla^*(h_c z^*)| = G_c |\nabla^* z^*|, \qquad (10)$$

where  $G_c$  is a characteristic gradient defined as

$$G_c \coloneqq \frac{h_c}{l_c} = \frac{U}{\sqrt{DK}}.$$
(11)

Our characteristic gradient  $G_c$  should not be confused with the critical slope  $S_c$  used in a non-linear diffusion law (e.g., Roering et al., 1999). Likewise, we can express topographic curvature as

$$\nabla^2 z = \frac{1}{l_c^2} \, \nabla^{*2}(h_c z^*) = \kappa_c \nabla^{*2} z^* \,, \tag{12}$$

where  $\kappa_c$  is a characteristic curvature defined as

$$\kappa_c \coloneqq \frac{h_c}{l_c^2} = \frac{G_c}{l_c} = U/D.$$
<sup>(13)</sup>

Our characteristic curvature should not be confused with the contour curvature, also denoted as  $\kappa_c$  elsewhere (e.g., Perron et al., 2012). The negative of U/D has been previously shown to describe the steady-state curvature of hilltops and drainage divides (e.g., Roering et al., 2007; Perron et al., 2009). Finally, given that areas scale as the square of lengths, we can express drainage area as (e.g., Perron et al. 2008, 2012)

$$A = l_c^2 A^* = A_c A^* , (14)$$

where  $A_c$  is a characteristic area, defined as

$$A_c \coloneqq l_c^2 = D/_K.$$
<sup>(15)</sup>

5

Substituting Eqs. (6), (8), (10), (12), and (14) into the governing equation (Eq. 1) yields

$$(h_c \partial z^*)/(t_c \partial t^*) = -K\sqrt{A_c A^*} G_c |\nabla^* z^*| + D \kappa_c \nabla^{*2} z^* + U$$

which can be simplified to the dimensionless form:

$$\frac{\partial z^*}{\partial t^*} = -\sqrt{A^*} |\nabla^* z^*| + {\nabla^*}^2 z^* + 1.$$
(16)

Equation (16) includes only dimensionless variables and no parameters. Because Eq. (16) has no parameters to be adjusted,

20

for a given set of boundary and initial conditions, it will have only one steady-state solution, which will be arrived at via only one path of evolution. This implies that all simulated landscapes with any parameters (but properly rescaled domains, boundary conditions, and initial conditions) will evolve as rescaled replicas of each other, because they can all be reduced to Eq. (16) through rescaling by the characteristic scales  $l_c$ ,  $h_c$ , and  $t_c$ . We explore this rescaling property in length in Sect. 3 and in Appendix B.

Alternative dimensionless forms of Eq. (1) can reveal properties of the LEM that are not revealed by Eq. (16). For example,

- Perron et al.'s (2008) Eq. (19) was derived using the domain half-width as a characteristic length and the steady-state 5 maximum relief as a characteristic height. In this way, Perron et al.'s (2008) dimensionless equation includes information about the domain size and the initial conditions (which influence the final relief; e.g., Howard, 1994); therefore, that equation highlights the dependence of its solutions on the domain size and the initial conditions. Perron et al.'s (2008) Eq. (19) is equivalent to our Eq. (16) if one can express the domain size and relief in terms of  $l_c$  and  $h_c$ , but the relationships
- between these two different pairs of scales will vary for different landscape configurations arising from different initial and 10 boundary conditions.

Likewise, Eq. (16) does not reveal that flow-routing algorithms, and thus LEM solutions, can be resolution-dependent if the channel width w is smaller than the mesh resolution  $\delta$  (e.g., Pelletier, 2010). This dependence can be minimized by including

- 15 the factor  $\delta/w$  in the diffusion term (e.g., Pelletier, 2010) or the factor  $w/\delta$  in the incision term (e.g., Perron et al., 2008). Equation (16) does not include such factors. However, the rescaling of domains (detailed in Sect. 3) guarantees that both w and  $\delta$  scale with  $l_{\alpha}$ ; this guarantees that the resolution-dependence of model solutions is consistent across rescaled landscapes.
- The fact that Eq. (16) includes no parameters has an additional important implication. One can use the factors that appear in 20 front of terms of a dimensionless equation to infer the relative importance of each term (e.g., Huntley, 1967). In the case of Eq. (16), all such factors are equal to one, which implies that none of the terms of this LEM (Eq. 1) is negligible everywhere across a landscape. In other words, each term may be dominant at some points of a landscape, depending on the local values of drainage area A, slope  $|\nabla z|$ , and curvature  $\nabla^2 z$ , even if it is negligible elsewhere. Therefore, none of the terms of Eq. (1)
- can be dropped purely on grounds of process dominance. 25

We can rewrite Eq. (16) in a form that reveals what controls the relative dominance of each process across a landscape. Specifically, the dimensionless quantities of Eq. (16) are equal to the ratio of the corresponding dimensional quantities over appropriate characteristic scales (Eqs. 8, 10, 12, and 14). Therefore, we can rewrite Eq. (16) as

$$\frac{\partial z/\partial t}{U} = -\frac{\sqrt{A}}{\sqrt{A_c}} \frac{|\nabla z|}{G_c} + \frac{\nabla^2 z}{\kappa_c} + 1.$$
(17)

- Equation (17) is exactly equivalent to Eq. (16) but helps illuminate different properties of the model. Specifically, Eq. (17) 30 shows that the relative contributions of each of the topographic properties A,  $|\nabla z|$ , and  $\nabla^2 z$  are equal to their ratios over the corresponding characteristic scales  $A_c$ ,  $G_c$ , and  $\kappa_c$ . The ratios in Eq. (17), or other, equivalent groupings of variables and parameters, could be defined as dimensionless numbers. Often, dimensional analyses use dimensionless numbers to express the relative contributions of processes. In the case of landscape evolution models, examples of such dimensionless numbers
- are uplift numbers (e.g., Whipple and Tucker, 1999) and Péclet numbers (e.g., Perron et al., 2008). Equation (17) expresses 35 how such dimensionless numbers emerge from the dimensionless governing equation.

#### 3 Scaling and similarity of landscapes

The dimensionless form of the governing equation (Eq. 16) implies that landscapes with any parameters, but properly rescaled boundary and initial conditions (see immediately below what we term as "proper" rescaling), will evolve in such a way that snapshots of these landscapes at properly rescaled moments in time will be (horizontally and vertically) rescaled

copies of each other. In other words, the evolution of these landscapes will obey temporal and geometric similarity. This, in turn, implies that such landscapes will reach geometrically similar steady states.

We consider domains, elevations, and time to be properly rescaled if they are equivalent when normalized by the characteristic length, height, and time scales  $l_c$ ,  $h_c$ , and  $t_c$ , respectively. For instance, let a landscape have parameters K, D,

and *U*, and a second landscape have parameters *K'*, *D'*, and *U'*. Variables and characteristic scales of the second landscape are primed to match the notation of its parameters. Domains are properly rescaled when pairs of points with dimensional coordinates (x, y) and (x', y') correspond to the same point with dimensionless coordinates  $(x^*, y^*)$ . Thus,  $x'^* = x^* \iff x'/l_c' = x/l_c \iff x' = (l_c'/l_c) x$ , (18 a)

and, likewise,

$$y' = (l_c'/l_c) y$$
. (18 b)

Likewise, elevations are properly rescaled when dimensional elevations z and z' at equivalent points (x, y) and (x', y') correspond to the same dimensionless elevation  $z^*$  at  $(x^*, y^*)$ , such that

$$z' = (h_c'/h_c) z$$
, (18 c)

and two moments in time t and t' are properly rescaled when they correspond to the same moment in dimensionless time  $t^*$  such that

$$t' = (t_c'/t_c) t$$
. (18 d)

We should point out that simulations of these two landscapes will reach geometrically similar steady states only if we rescale the threshold  $\varepsilon$  in the steady-state criterion of Eq. (2). Specifically, we assume that the two landscapes have reached

the threshold ε in the steady-state criterion of Eq. (2). Specifically, we assume that the two landscapes have reached numerical steady states that satisfy the criteria |∂z/∂t | ≤ ε and |∂z'/∂t' | ≤ ε'. Using Eq. (8) we see that in the dimensionless coordinate system of Eq. (16) these criteria become | U(∂z\*/∂t\*) | ≤ ε ⇔ | (∂z\*/∂t\*) | ≤ ε/U ≔ ε\* and, likewise, | U'(∂z\*/∂t\*) | ≤ ε' ⇔ | (∂z\*/∂t\*) | ≤ ε' ⇔ | (∂z\*/∂t\*) | ≤ ε' ⇔ | (∂z\*/∂t\*) | ≤ ε'. If the two numerical steady states are geometrically similar, then they must be satisfying the same dimensionless criterion, i.e., ε'\* = ε\*, which leads to a steady-state threshold rescaling formula:

$$\varepsilon'/U' = \varepsilon/U \Leftrightarrow \varepsilon' = (U'/U)\varepsilon.$$
 (19)

In the following subsections we use a numerical model to demonstrate the temporal and geometric similarity of rescaled landscapes that is implied by the dimensionless governing equation (Eq. 16). In addition, in Appendix B we outline a simple analytical proof of this similarity property. That proof suggests that rescaling works only if we rescale initial conditions

30 (elevations) by  $h_c$ . This implies that rescaling works only if  $l_c$  and  $h_c$  are defined separately, i.e., only if we assume distinct dimensions for lengths and heights.

### 3.1 Numerical demonstration

## 3.1.1 Model set-up

We used the Channel-Hillslope Integrated Landscape Development (CHILD) model (Tucker et al., 2001) to numerically demonstrate the similarity property revealed by the dimensionless governing equation (Eq. 16). We chose CHILD due to its

- wide use by the geomorphologic community, and due to the fact that it uses triangular irregular networks (TIN), which avoid the geometric bias of regular grids (e.g., Braun and Sambridge, 1997). We selected CHILD modules and parameters such that it would simulate Eq. (1). Specifically, we selected CHILD's detachment-limited incision module, with constant, uniform precipitation, along with linear diffusion and uniform uplift (see Tucker et al., 2001 and Tucker, 2010 for definitions of CHILD's assumptions, modules, and parameters). In Appendix C we present in more detail how we set up our CHILD
- simulations and how we retrieved our results from CHILD's output files.

We ran simulations using multiple combinations of the model parameters *K*, *D*, and *U*. We chose baseline parameter values of  $K = 10^{-6} a^{-1}$ ,  $D = 10^{-2} m^2 a^{-1}$ , and  $U = 10^{-4} m a^{-1}$ , which are typical in the literature (e.g., Perron et al., 2008; Tucker, 2009; Clubb et al., 2016), and we varied each parameter by two orders of magnitude around its baseline value, in a total of 24 perameter combinations.

total of 34 parameter combinations.

We applied these parameter combinations on rescaled copies of two random, dimensional TINs. In Appendix C1.2 we describe how we prepared the rescaled TINs. Domain size was 200  $l_c$  by 400  $l_c$  with an average mesh edge of 0.8  $l_c$ , resulting in approximately 150,000 TIN vertices. Initial elevations were a uniform white noise, ranging between 0 and 0.1  $h_c$ . For each

simulation we calculated  $l_c$  and  $h_c$  according to Eqs. (3) and (4), respectively. We rescaled the horizontal TIN coordinates and initial elevations according to Eqs. (18 a–c).

Simulation time step lengths were not explicitly rescaled. Rather, we defined simulation time step lengths using Courant-Friedrichs-Lewy criteria (Refice et al, 2012) as described in Appendix C (Eq. C3). As seen in Eq. (C4), it turns out that the resulting time step lengths were in effect rescaled by  $t_c$  due to the dependence of the Courant-Friedrichs-Lewy criteria on

rescaled variables.

We ran simulations until they reached numerical steady states, which we defined using rescaled steady-state thresholds according to Eq. (19). We compared the resulting landscapes during their evolution and at their steady states.

## 30 3.1.2 Numerical results

The numerical results confirmed the rescaling properties of the dimensionless governing equation (Eq. 16). Simulations which were run on rescaled versions of the same random TIN evolved similarly in space and time. Specifically, at time steps rescaled by  $t_c$  (Eq. 18 d), elevations of corresponding points across simulations could be rescaled by  $h_c$  (Eq. 18 c) Furthermore, if steady-state criteria were rescaled according to Eq. (19), then simulations reached geometrically similar steady states.

In Figs. 1 and 2 we present steady-state results for three landscapes (our baseline and two alternatives), and in Figs. 3–5 we present transient results during their evolution. These figures illustrate the geometric and temporal similarity of these

landscapes. The parameter combinations of these three landscapes are a subset of all the combinations that we used; their values can be seen in Table 2. We are presenting these specific combinations for demonstration purposes as they lead to wide ranges of  $l_c$  and  $h_c$ . However, all the other parameter combinations that we tested also yielded landscapes that exhibited temporal and geometric similarity.

5

25

In Fig. 1 we show steady-state shaded relief maps, and in Fig. 2 we show elevation maps and transects. In both figures  $l_c$  and  $h_c$  increase from left to right. We vary  $l_c$  and  $h_c$  separately; thus, the characteristic gradient  $G_c$  does not vary monotonically from left to right. In Fig. 2, the thick black lines in the top panels mark transects corresponding to the elevation profiles in the bottom panels, and pass through the highest peaks of the simulated landscapes. The coloring and labeling of Figs. 1 and 2

- 10 highlight both the large differences of scale and the geometric similarity of the three rescaled landscapes. For comparison, lengths and elevations on axes and colorbars are shown both in units of km or m using bold fonts, and in units of  $l_c$  or  $h_c$ using normal fonts. Note that the  $l_c$  and  $h_c$  values for different landscapes are different as they depend on the model parameters. Color scales of elevation maps in Fig. 2 are rescaled by  $h_c$  to assist with comparing the elevations of features.
- 15 In the shaded relief maps of Fig. 1, the spatial pattern of ridges and valleys is identical across the three landscapes, illustrating their horizontal geometric similarity, although their shaded relief contrast varies, reflecting their different characteristic gradients  $G_{c}$ . Likewise, in the elevation maps of Fig. 2, the spatial pattern of colors is identical across the three landscapes. This illustrates that the three landscapes are geometrically similar both horizontally and vertically, because the color scales are rescaled by  $h_c$ . Finally, the horizontal and vertical geometric similarity of the three landscapes is illustrated 20

also by the shapes of the transects of Fig. 2.

The geometric similarity of the three steady-state landscapes is exact, not just visually convincing. Our domain rescaling procedure does not affect the point IDs of the TIN vertices, so we can directly compare corresponding points using their IDs. In these simulated landscapes, the maximum absolute difference of dimensionless elevations  $z^*$  of corresponding points was less than  $10^{-9}$  units of  $h_c$ .

Figures 3–5 show shaded relief maps, elevation maps, and transects for four snapshots in time during the evolution of the three landscapes. Each column shows snapshots of the three landscapes that correspond to the same moment in dimensionless time (but different moments in dimensional time), with time increasing from left to right. Each row shows the

- evolution of one landscape, with one set of model parameters. As in Figures 1 and 2, lengths and elevations on axes and colorbars in units of km or m are shown in bold, and the corresponding values in units of  $l_c$  or  $h_c$  are shown in normal font for comparison. Labels above each snapshot show time in millions of years (in bold fonts) and in units of  $t_c$  (in normal fonts). Color scales of elevation maps in Fig. 4 are rescaled by  $h_c$  to assist with comparing the elevations of features. For each landscape, we use one color scale that remains constant in time to highlight how relief evolves. Each landscape's color scale
- is set to match the highest elevation among the four snapshots. We use different color scales for different landscapes and rescale them by  $h_c$  to facilitate comparison of features across landscapes. Visual comparison shows the three landscapes to be temporally similar, since at the same moments in properly rescaled time they are in geometrically similar transient states. Note that  $l_c$  and  $h_c$  are different across landscapes, but for each landscape, they are constant in time. Note also that the snapshots that we present are not equally spaced in time, because these landscapes evolve rapidly at first, and much more
- slowly later.

As in the case of steady-state landscapes, the temporal and geometric similarity of the three evolving landscapes is exact, not just visually convincing. Throughout the evolution of the three landscapes, the maximum absolute difference of dimensionless elevations  $z^*$  of corresponding points at corresponding moments of dimensionless time is less than  $10^{-7}$  units of  $h_c$ .

#### 3.2 Implications of temporal and geometric similarity

5

25

## 3.2.1 Deducing how model parameters control landscape metrics

The geometric similarity of rescaled landscapes implies that all horizontal coordinates (x, y) and elevations z, and thus all lengths and heights, will be rescaled by the characteristic length and height scales  $l_c$  and  $h_c$ , respectively, according to

- Eqs. (18 a–c). Likewise, the temporal similarity of the evolution of rescaled landscapes implies that all time intervals will be rescaled by the characteristic time scale  $t_c$  according to Eq. (18 d). Thus, any variables that combine the dimensions of length L, height H, and/or time T will be rescaled by the corresponding combinations of  $l_c$ ,  $h_c$ , and  $t_c$ . For example, as we showed in Sect. 2.2, drainage areas, slopes, curvatures, and rates of elevation change are rescaled by the characteristic area  $A_c = l_c^2$ , characteristic gradient  $G_c = h_c/l_c$ , characteristic curvature  $\kappa_c = h_c/l_c^2$ , and uplift rate  $U = h_c/t_c$  (we showed that U can
- be viewed as a characteristic rate of elevation change; Eq. 8).

The characteristic length, height, and time scales  $l_c$ ,  $h_c$ , and  $t_c$  depend only on the model parameters *K*, *D*, and *U* (Eqs. 3–5). Therefore, we can infer how any variable scales with *K*, *D*, and *U* based on how its corresponding characteristic scale combines  $l_c$ ,  $h_c$ , and  $t_c$  (which we can infer from how the variable's dimensions combine L, H, and T). For example, from the

20 definitions of  $l_c$ ,  $G_c$ ,  $h_c$ ,  $t_c$ ,  $A_c$ ,  $\kappa_c$ , and U (Eqs. 3, 11, 4, 5, 15, 13, 8) we infer that if we change the incision coefficient by a factor k, i.e., if we change it from K to kK, then all distances and slopes will change by  $1/\sqrt{k}$ , all reliefs, durations, and drainage areas will change by 1/k, and all (Laplacian) curvatures and rates of elevation change will remain the same.

As an additional example, the ratio of the characteristic length  $l_c$  to the characteristic time  $t_c$  defines a characteristic horizontal velocity

$$u_c \coloneqq \frac{l_c}{t_c} = \sqrt{DK} \,. \tag{20}$$

We can, thus, deduce that horizontal velocities of drainage divide migration must scale by  $u_c$ , i.e., by  $\sqrt{DK}$ . This does not imply that all drainage divides move with velocity  $u_c$ . Rather, it implies that any formula describing drainage divide migration must scale as  $\sqrt{DK}$ , and cannot include any other terms that depend on the model parameters *K*, *D*, and *U*. Such a formula will also depend on factors that vary locally across the landscape (such as, for example, divide asymmetry), and

30 which must be derived separately for specific cases; they cannot be derived by scaling considerations. These principles provide a plausibility check for theoretical predictions of drainage divide migration in landscapes that follow Eq. (1). The same general approach may also be applicable in landscapes that follow other governing equations, if one can define a characteristic velocity (which may scale differently than the example shown in Eq. 20).

#### 3.2.2 Improving modeling efficiency

The temporal and geometric similarity of our rescaled simulations (Sect. 3.1) implies that we can explore the entire K, D, and U parameter space by rescaling a single simulation. For instance, if we are interested in how the slope–area curve depends on these three parameters, we can run one simulation with one combination of K, D, and U and plot its slope–area curve. To

5 obtain the slope–area curve for any other combination of parameters K', D', and U', we can simply rescale slopes by the characteristic gradient and drainage areas by the characteristic area, i.e., we can multiply slopes by  $(U'/\sqrt{D'K'})/(U/\sqrt{DK})$  and drainage areas by (D'/K')/(D/K) (see Eqs. 11 and 15). The resulting rescaled drainage areas and slopes will be exactly equal to the drainage areas and slopes of a simulation with parameters K', D', and U', and with domain size, resolution, and initial conditions that are rescaled as described in Sect. 3.1.

Exploring a parameter space by rescaling can be orders of magnitude more efficient than running multiple simulations for multiple parameter combinations. For example, consider a numerical experiment exploring 10 values for each of the three parameters K, D, and U, in all possible combinations. Exploring this parameter space by brute force would require 1000 simulations, versus just one simulation with the rescaling approach.

Inferring the results of a simulation by rescaling the results of another simulation assumes that the sizes of the simulation domains are equal in units of characteristic length  $l_c$  (i.e., domain sizes are rescaled by  $l_c$ ); this may often be physically unrealistic. For example, if the simulation domain represents an island, increasing  $l_c$  (e.g., by increasing the diffusion coefficient *D*) will make the island less dissected (i.e., will increase the spacing of valleys), but will not make the island

- bigger as would be required to keep the domain size constant in units of *l<sub>c</sub>*. Consequently, the original island will look rougher than the island with increased *l<sub>c</sub>*, and the two islands, overall, will not be geometrically similar, even in a statistical sense. Locally, however, features that are much smaller than both islands, and sufficiently far from the coastlines, will be insensitive to whether the coastlines are rescaled or not; thus, these features may be statistically similar (even if their exact spatial patterns differ). Therefore, our rescaling approach may give us insight into how model parameters control the
  behavior of sufficiently small features, recordless of whether domain reception is assumed.
- behavior of sufficiently small features, regardless of whether domain rescaling is assumed.

However, if we vary the model parameters such that the features of interest are no longer small with respect to the domain, then these features will be influenced by boundary effects and may not be able to express their intrinsic shapes or behaviors, i.e., the shapes or behaviors that they would have if they were small relative to the domain. On the other hand, if we vary the

30 parameters such that features of interest are not sufficiently large with respect to the resolution, then these features may be influenced by resolution effects, because they may be insufficiently resolved. In both of these cases, we can no longer reasonably assume that we can study the features of interest with a rescaling approach.

To be able to assess which combinations of domain sizes and resolutions, and model parameters K, D, and U could result in boundary or resolution effects, one should consider domain sizes and resolutions in units of characteristic length  $l_c$ . For example, the regime transition from dominant diffusion to dominant incision occurs at length scales of the order of  $l_c$  as shown by Perron et al. (2008, 2009); see also Sect. 4.2.2 below. Thus, if we want to study this regime transition, we should vary the model parameters such that  $l_c$  remains sufficiently small compared to the domain size and sufficiently large compared to the resolution. How small is sufficiently small (and, likewise, how large is sufficiently large) will not be known

a priori. However, if, within a range of values of  $l_c$ , a feature's properties and behavior scale according to  $l_c$ ,  $h_c$ , and  $t_c$  as

described in Sect. 3.2.1 (e.g., if drainage divide migration velocity follows Eq. (20)), then we can infer a posteriori that it can be studied with a rescaling approach over that range of  $l_c$ .

In general, one should consider all the specifications of simulations not only in units of meters and years, but also in units of

$l_c$ ,  $h_c$ , and  $t_c$  (e.g., amplitudes of initial conditions in units of  $h_c$ , rates of elevation change in units of  $h_c t_c^{-1}$ , etc.). Likewise, we recommend converting simulation results into units of  $l_c$ ,  $h_c$ , and  $t_c$  (e.g., drainage densities in units of  $l_c^{-1}$ , drainage areas of valley heads in units of  $l_c^2$ , response times in units of  $t_c$ , etc.). This can be helpful in comparing seemingly disparate model results, and identifying which metrics and features can be studied with a rescaling approach.

## 4 Interpretations of the characteristic scales

The values of the characteristic scales depend on the relative magnitudes of the model parameters K, D, and U, and thus on the relative strengths of incision, diffusion, and uplift. In the present section, we show that the characteristic scales can link the relative strengths of these processes with topographic properties of the landscape. Thus they can aid the study of process competition and regime transitions.

#### 4.1 Height scales

#### 15 4.1.1 Using height scales to quantify the vertical influence of incision, diffusion, and uplift

The incision, diffusion, and uplift terms of the governing equation (Eq. 1) give the rates of change of elevation due to the respective processes. We can scale these rates using the characteristic time  $t_c$  along with the characteristic height  $h_c$  and two additional height scales that we introduce in this section.

- The definitions of the characteristic height ( $h_c = U/K$ ; Eq. 4) and the characteristic time ( $t_c = 1/K$ ; Eq. 5) show that  $h_c$  is the elevation uplifted per unit of  $t_c$ , i.e.,  $h_c = U t_c$ . Therefore, we can view  $h_c$  as a scale that measures the contribution of uplift to elevation change per unit of  $t_c$ . We extend this notion to the incision and diffusion terms of the governing equation (Eq. 1), and define an incision height scale as the erosion due to incision per unit of  $t_c$ ,  $h_I \coloneqq K\sqrt{A}|\nabla z| t_c = \sqrt{A}|\nabla z|$ , (21) and a diffusion height scale as the elevation change due to diffusion per unit of  $t_c$ ,  $h_D \coloneqq D\nabla^2 z t_c = l_c^2 \nabla^2 z$ . (22)
- Intuitive interpretations of  $h_l$ ,  $h_D$ , and  $h_c$  are schematically illustrated in Fig. (6), which is described in more detail in the following subsection (Sect. 4.1.2). The incision height is defined as a positive quantity, but because it measures erosion we should remember that it is pointing in a downward direction. The diffusion height is negative for erosive diffusion and positive for depositional diffusion. We observe that, for the case of Eq. (1), which has drainage area and slope exponents m = 0.5 and n = 1,  $h_l$  is equal to the steepness index, defined as  $k_s = A^{m/n} |\nabla z|$  (e.g., Whipple, 2001). For slope exponents
- $n \neq 1$ , however,  $k_s$  and  $h_l$  are not equal;  $k_s$  is proportional to  $h_l^{1/n}$  (Eq. A19).

In geometrically similar landscapes, the incision and diffusion heights of corresponding points will be rescaled by the characteristic height  $h_c$  in the same way as all other variables with dimensions of height (e.g., elevations, reliefs, etc.). Specifically, assume that two geometrically similar landscapes have parameters K, D, and U, and K', D', and U' (and thus have characteristic scales  $l_c$ ,  $h_c$ , and  $t_c$ ,  $h_c'$ , and  $t_c'$ ; note that primed variables refer to the second landscape, whose

parameters are also primed). As we explain in Sect. 3 and Appendix B, corresponding points in these landscapes, i.e., points with coordinates such that  $x'/l_c' = x/l_c$  and  $y'/l_c' = y/l_c$  (Eqs. 18 a, b), will have drainage areas and slopes such that  $\sqrt{A'}/l_c' = \sqrt{A}/l_c$  and  $|\nabla'z'|/G_c' = |\nabla z|/G_c$ . Therefore, they will have incision heights that are related according to:  $h_I'/h_c' = \sqrt{A'}|\nabla'z'|/h_c' = (l_c'/l_c)\sqrt{A}$  ( $G_c'/G_c$ ) $|\nabla z|/h_c' = \sqrt{A}|\nabla z|/h_c = h_I/h_c$ . (23 a) Likewise, one can show that they will have diffusion heights that are related according to:

$$h_D'/h_c' = h_D/h_c \,.$$

(23 b)

Equations (23 a, b) are examples of the ability of a characteristic scale to rescale variables with which it has the same

dimensions – in this case the characteristic height  $h_c$  rescales  $h_I$  and  $h_D$  even though they are not physical heights (they just have dimensions of height).

Note that Eq. (23 a) shows that, if we define a dimensionless incision height  $h_l^*$  as the ratio of the incision height  $h_l$  to the characteristic height  $h_c$ , then it will be equal to the dimensionless incision terms of Eqs. (16) and (17), i.e.,

$$h_I^* \coloneqq h_I / h_c = \left(\sqrt{A} |\nabla z|\right) / \left(\sqrt{A_c} \ G_c\right) = \sqrt{A^*} |\nabla^* z^*| .$$
(24 a)

15 Likewise, one can show that an analogously defined dimensionless diffusion height  $h_D^*$  will be equal to the dimensionless diffusion terms of Eqs. (16) and (17), i.e.,

$$h_D^* \coloneqq h_D / h_c = \nabla^2 z / \kappa_c = \nabla^{*2} z^* .$$
(24 b)

Equations (24 a, b) highlight that the three terms of the dimensionless Eqs. (16) and (17) quantify the relative contributions of incision, diffusion, and uplift to elevation change.

# 4.1.2 Properties of the incision, diffusion, and characteristic height scales $h_I$ , $h_D$ , and $h_c$

We can express the governing equation Eq. (1) in terms of the incision, diffusion, and characteristic height scales  $h_l$ ,  $h_D$ , and  $h_c$  if we multiply it by the characteristic time  $t_c$ :

$$(\partial z/\partial t) t_c = -h_I + h_D + h_c.$$
(25)  
In steady state  $(\partial z/\partial t = 0)$ , Eq. (25) yields  
 $0 = -h_I + h_D + h_c$ , for  $\partial z/\partial t = 0$ ,
(26)

which we can manipulate in various ways to reveal useful properties of the three height scales  $h_{I}$ ,  $h_{D}$ , and  $h_{c}$ .

First, we focus on points that have zero curvature ( $\nabla^2 z = 0$ ). At these points, the net effect of diffusion on elevation is zero; thus, incision and uplift must be in balance with each other in steady state. Setting  $h_D = 0$  in Eq. (26), we can mathematically express the incision–uplift balance at these points in two equivalent ways:

$$h_I = h_c \iff \sqrt{A} |\nabla z| = U/K$$
, for  $\partial z/\partial t = 0$ ,  $\nabla^2 z = 0$ . (27)

These expressions show that the characteristic height  $h_c$  determines the steady-state value of the incision height  $h_I$  at points with zero curvature so that incision and uplift are in balance with each other (since diffusion has a zero net contribution to

30 elevation change at these points). Points with zero curvature represent a regime transition between concave-down hillslopes, characterized by net erosion by diffusive transport, and concave-up valleys, characterized by net deposition by diffusive

transport (e.g., Howard, 1994). Thus, a notable implication of Eq. (27) is that points with  $h_I = h_c$  will map out this important topography- and process-related regime transition in steady state. Equation (27) is reminiscent of a bedrock river steady-state slope–area relation, and we discuss the similarities and differences between them in Sect. 4.1.3.

- 5 Second, we focus on drainage divides, where the drainage area A is zero and there is no incision. At drainage divides, diffusion and uplift must be in balance with each other in steady state. Setting  $h_I = 0$  in Eq. (26), we express the diffusion– uplift balance on drainage divides as  $h_D = -h_c$ . Substituting the definitions of  $h_D$  and the characteristic curvature  $\kappa_c$ , and rearranging, yields the steady-state value of drainage divide curvature (e.g., Roering et al., 2007; Perron et al., 2009):  $\nabla^2 z = -h_c/l_c^2 = -\kappa_c = -U/D$ , for  $\partial z/\partial t = 0$ , A = 0. (28) This relation shows that we can view the characteristic height  $h_c$  and length  $l_c$  as two distinct components (one vertical, the
- 10 other horizontal) that jointly determine the steady-state curvature of drainage divides so that diffusion and uplift are in balance.

Equations (27) and (28) refer to special points where incision or diffusion is zero. These are special cases of a general steady-state property of  $h_c$  that is valid at all points; rearranging Eq. (26) yields

$$h_c = h_I - h_D$$
, for  $\partial z / \partial t = 0$ , (29)

which shows that the difference  $h_I - h_D$  is constant and equal to  $h_c$  across steady-state landscapes.

Figure 6 schematically illustrates how the incision, diffusion, and characteristic height scales  $h_I$ ,  $h_D$ , and  $h_c$  vary along a steady-state profile. In Fig. 6 b, the green line shows a steady-state profile that traces a flow path from the drainage divide to a point in a valley. Subtracting  $h_I$  from the elevations along the profile, or adding  $h_D$  or  $h_c$  to them, yield three gray lines, one

- solid, one dashed, and one dotted, that respectively show the individual contributions of incision, diffusion, and uplift to elevation change per unit of  $t_c$ . (These contributions are equivalent to how the profile would change if only incision, diffusion, or uplift operated on it, at their equilibrium rates, for one unit of  $t_c$ .) The three contributions must sum to zero at all points along this equilibrium profile. Whereas Fig. 6 b shows elevations and elevation changes, Fig. 6 a shows the values of  $h_l$ ,  $h_D$ , and  $h_c$  along the profile, using black lines that have the same shapes as the corresponding gray lines of Fig. 6 b.
- Figure 6 a schematically illustrates the relationships described by Eqs. (27), (28), and (29). Specifically, at the divide (point P<sub>1</sub>),  $h_I$  is zero and  $h_c$  and  $h_D$  are equal and opposite; at the point of zero curvature (point P<sub>2</sub>; also shown magnified in Fig. 6 b),  $h_D$  is zero and  $h_I$  equals  $h_c$ ; and over the entire profile,  $h_c = h_I - h_D$  (the spacing between the dashed and solid black lines is constant and equal to  $h_c$ ).

## 30 Substituting the definitions of $h_I$ and $h_D$ (Eqs. 21, 22) into Eq. (29) yields

$$h_c = \sqrt{A} |\nabla z| - l_c^2 \nabla^2 z$$
, for  $\partial z / \partial t = 0$ ,

which shows that the constant difference  $h_I - h_D$  implies that, in steady state, drainage areas *A*, slopes  $|\nabla z|$ , and curvatures  $\nabla^2 z$  are constrained by a relationship that is constant across the landscape and is parameterized by  $h_c$  (along with  $l_c$ ). In this sense, we can interpret the characteristic height  $h_c$  as a parameter that constrains the steady-state values of the drainage area *A*, slope  $|\nabla z|$ , and curvature  $\nabla^2 z$  across the landscape so that incision, diffusion, and uplift are in balance.

(30)

35

For drainage area and slope exponents *m* and *n* such that  $2m \neq n$  (Eq. A1),  $h_c$  depends on all three parameters *K*, *D*, and *U* (Eq. A9). Therefore, we should not interpret the characteristic height  $h_c$  as a scale that expresses the relative strength of uplift

versus incision (as the definition  $h_c = U/K$  (Eq. 4) may seem to suggest), but rather interpret it as expressing the relative strengths of all three processes. Thus, the aforementioned interpretation that  $h_c$  constrains steady-state topography so that all three processes are in balance is in line with the definition of  $h_c$  for generic exponents *m* and *n*.

5 Given that Eq. (30) can be rewritten as

$$\sqrt{A}|\nabla z| = l_c^2 \nabla^2 z + h_c, \quad \text{for } \partial z/\partial t = 0, \tag{31}$$

if we plot the product  $\sqrt{A}|\nabla z|$  versus the curvature  $\nabla^2 z$  we can graphically illustrate how  $h_c$  and  $l_c$  constrain A,  $|\nabla z|$ , and  $\nabla^2 z$  in a steady-state landscape. Additionally, we can graphically illustrate the special cases on points with zero curvature and on drainage divides described by Eqs. (27) and (28), respectively. We show such a plot in Fig. 7 using data from the simulated equilibrium landscape B (shown in Figs. 1 and 2). Figure 7 shows that  $\sqrt{A}|\nabla z|$  versus  $\nabla^2 z$  plot on a straight line, as

10 demanded by Eq. (31). In Fig. 7, the vertical dashed lines show the values  $\nabla^2 z = 0$  and  $\nabla^2 z = \kappa_c$ ; the horizontal dashed lines show the values  $\sqrt{A}|\nabla z| = h_c$  and  $\sqrt{A}|\nabla z| = 2h_c$ . Therefore, these dashed lines illustrate that the straight line on which the data points plot has an intercept equal to the characteristic height  $h_c$ , and a slope equal to  $h_c/\kappa_c$ , i.e., equal to  $l_c^2$ , the square of the characteristic length, as demanded by Eq. (31).

## 4.1.3 Quantitative predictions

30

- 15 Equations (30) and (31), the relationships that constrain steady state topography, are testable predictions. They imply that we can test whether the governing equation Eq. (1) describes a given, presumably steady-state, real-world landscape by plotting estimates of the product  $\sqrt{A}|\nabla z|$  versus estimates of the curvature  $\nabla^2 z$  from across this landscape. These estimates should plot on a straight line, as shown in Fig. 7 for our simulated landscape. (Note that our analysis only shows that Eqs. (30) and (31) must hold for a steady-state landscape governed by Eq. (1); we have not shown that a landscape that conforms to
- 20 Eqs. (30) and (31) is necessarily governed by Eq. (1), or in steady state. Although it seems unlikely that data from a nonsteady-state landscape that follows different geomorphic transport laws would happen to plot according to Eqs. (30) and (31), this premise should be tested using numerical experiments.)

Equation (31) can be used to estimate model parameters. Specifically, we can estimate  $l_c^2$  (i.e., the D/K ratio) from the slope of plots of the product  $\sqrt{A}|\nabla z|$  versus the curvature  $\nabla^2 z$ , and estimate  $h_c$  (i.e., the U/K ratio) from the intercept of these plots. Alternatively, given that  $\kappa_c = h_c / l_c^2$  (Eq. 13), we can rewrite Eq. (31) as

$$\sqrt{A}|\nabla z| = \frac{h_c}{\kappa_c} \nabla^2 z + h_c = h_c \frac{\nabla^2 z + \kappa_c}{\kappa_c},$$
(32)

and estimate  $h_c$  as the slope of plots of the product  $\sqrt{A}|\nabla z|$  versus the quantity  $(\nabla^2 z + \kappa_c)/\kappa_c$ . To apply Eq. (32) to realworld data we would need to estimate  $\kappa_c$ , e.g., as discussed above from steady-state drainage-divide curvature (e.g., Perron et al., 2009). Note that Eqs. (30)–(32) are equivalent to Eq. (5) of Perron et al. (2009), which they used to estimate  $l_c$  and which they recognized as a test for model validity. For instance, dividing Eq. (32) by  $\kappa_c$  yields Eq. (5) of Perron et al. (2009).

Another equation that could hypothetically be used to estimate model parameters is Eq. (27), which is reminiscent of the steepness-index formula  $A^{m/n}|\nabla z| = (U/K)^{1/n}$  that has been used to estimate model parameters from steady-state profiles of bedrock rivers (e.g., Sklar and Dietrich, 1998; Whipple, 2001). Although the two equations are similar in form, they apply

35 to different points on the landscape. Equation (27) describes points of zero curvature, where diffusive transport is negligible,

whereas the steepness-index formula is applied to bedrock rivers, where curvature is clearly not zero but diffusive transport is nonetheless assumed to be zero (and thus Eq. (1) does not apply). To estimate  $h_c$ , Eq. (27) can only use the relatively few points with zero curvature, whereas Eq. (32) can use data from the whole landscape and, thus, would presumably yield more robust estimates of  $h_c$ .

#### 5 4.1.4 Valley definition

Valleys have been defined as areas where the quantity  $A(|\nabla z|)^2$  exceeds some threshold value (e.g., Montgomery and Dietrich, 1992; Orlandini et al., 2011; Clubb et al., 2014). This quantity is equal to  $(h_I)^2$ , the square of the incision height. Montgomery and Dietrich (1992) used thresholds of  $A(|\nabla z|)^2$  as criteria for defining channels and valleys, concluding that channelization and valley incision are controlled by the same topographic properties. Other authors have used curvature  $\nabla^2 z$ 

- to define valleys; specifically, valleys have been defined as regions with curvature above some threshold, i.e.,  $\nabla^2 z \ge \kappa_{thr}$ , where the threshold curvature  $\kappa_{thr}$  is assumed to be zero (e.g., Howard, 1994) or a small positive value (e.g., Lashermes et al., 2007; Pelletier, 2013). Here we demonstrate that these seemingly disparate criteria are closely related in steady-state landscapes that follow Eq. (1).
- Equations (26) and (31) can be combined as

$$h_{I} = \sqrt{A} |\nabla z| = l_{c}^{2} \nabla^{2} z + h_{c}, \quad \text{for } \partial z / \partial t = 0,$$
(33)

which shows that, in steady state,  $\sqrt{A}|\nabla z|$  – the square root of Montgomery and Dietrich's valley criterion – is linearly related to topographic curvature  $\nabla^2 z$ . Equation (33) shows that points with  $\nabla^2 z \ge \kappa_{thr}$  will be identical to points whose incision heights  $h_I = \sqrt{A}|\nabla z|$  exceed a corresponding threshold value  $h_{thr}$ . Figure 8 graphically illustrates the property of  $h_I$  to define valleys. It shows that coloring the simulated equilibrium landscape B (shown in Figs. 1 and 2) using the  $h_I$  value of

20 each pixel reveals a dendritic pattern. Given that  $h_I$  and  $\nabla^2 z$  are linearly related, they have identical spatial patterns; thus, the dendritic pattern of Fig. 8 shows the valley network defined by either criterion.

Given that one is a linear function of the other, are there practical reasons to prefer  $A(|\nabla z|)^2$  or  $\nabla^2 z$  as a valley definition criterion? The first criterion requires computing drainage areas to each point on the landscape, which can be computationally tedious, whereas the second criterion requires estimating curvatures, which can be sensitive to topographic noise. Note, however, that thresholds of the quantity  $A(|\nabla z|)^2$  correspond to curvature thresholds only if 2m = n. In the more general case of  $2m \neq n$  (Eq. A1), curvature thresholds will correspond to thresholds of the quantity  $A(|\nabla z|)^{n/m}$  (Eq. A26), rather

than  $A(|\nabla z|)^2$  (e.g., Ijjasz-Vasquez and Bras, 1995).

## 4.1.5 Divide migration dynamics

- Equation (33) holds only in steady state; its counterpart in transient states would be  $(\partial z/\partial t)t_c + h_I = l_c^2 \nabla^2 z + h_c$ , as implied by Eq. (25). In transient states, the term  $(\partial z/\partial t)t_c$  varies across the landscape. Thus the incision height  $h_I$  is not linearly related with curvature  $\nabla^2 z$ , but nonetheless it remains a useful quantity. For instance, it reproduces Whipple et al.'s (2017) finding that erosion rate differences across drainage divides can predict the direction of divide migration. In Fig. 9 we show four snapshots of the simulated evolving landscape B (shown in Figs. 3 and 4), which we colored using the  $h_I$  value of
- ach pixel. This coloring revealed that  $h_I$ 's spatial distribution followed dendritic patterns. Furthermore, it revealed that dendritic patterns across migrating drainage divides had different colors, i.e., different  $h_I$  values. Finally, it revealed that

drainage basins overwhelmingly tended to expand at the expense of neighbors with dendritic patterns with relatively lower  $h_I$  values. Figure 9 suggests that  $h_I$  predicts the direction of divide migration. This property characterized the entire evolution of landscape B, but was more evident during the early phases from which we chose the four snapshots of Fig. 9. For the case of Eq. (1),  $h_I$  is equal to the steepness index  $k_s$ , which was one of the metrics that Whipple et al. (2017) used to measure erosion rates; thus Fig. 9 also illustrates the use of  $k_s$  to predict divide migration dynamics.

#### 4.2 Length and time scales

Perron et al. (2008, 2009, 2012) expressed the competition between diffusion, which smooths landscapes, and incision, which dissects them, in terms of a Péclet number  $Pe = Kl^2/D$ , where *l* is a length scale that characterizes landscape features of interest. Their analysis implied that incision and diffusion should be equally effective when the Péclet number is roughly

- equal to one, and thus the length scale is roughly equal to  $\sqrt{D/K}$ , i.e., to the characteristic length  $l_c$ . They showed that distances between equally spaced valleys scale with  $l_c$ , while drainage areas of first-order valley heads and of second- to first-order valley branching scale with  $l_c^2$ , i.e., with the characteristic area  $A_c$ . Here, we introduce a related, but different, definition of the Péclet number and explore its implications. One such implication is that we can use this Péclet number to interpret the characteristic length and time scales  $l_c$  and  $t_c$  as scales that characterize a transition between regimes of
- dominant diffusion and dominant incision.

## 4.2.1 Quantifying the horizontal influence of incision and diffusion

To examine the properties of *l<sub>c</sub>* we focus on the horizontal effects of incision, diffusion, and uplift. This approach is in line with our goal to study horizontal and vertical scaling separately, and with the assumption that the dimensions of length and height are distinct. Equation (1) explicitly defines rates of elevation change, i.e., effects of processes in the vertical direction.
However, the incision and diffusion terms have additional, implicit horizontal effects. Specifically, incision advects topographic perturbations, such as knickpoints, and diffusion smooths them (e.g., Whipple and Tucker, 1999; Perron et al., 2008). We can quantify the strength of these effects using time scales that characterize incision and diffusion.

The incision term of Eq. (1) has the form of a kinematic wave, and advects perturbations at a rate equal to the kinematic 25 wave celerity  $c = K\sqrt{A}$  (e.g., Whipple and Tucker, 1999). The time needed to advect perturbations over some distance *l* gives an appropriate measure of incision's horizontal influence; thus, we define an incision time scale as:

$$t_I \coloneqq \frac{l}{K\sqrt{A}}.\tag{34}$$

A small  $t_l$  value corresponds to a strong horizontal influence of incision (the stronger the incision, the less time is needed for advection over a given distance l).

The diffusion term of Eq. (1) smooths elevation differences by redistributing them over an expanding region of neighboring points. For example, an elevation difference that is initially concentrated at a single point will evolve as a Gaussian function, centered around this point and with a standard deviation that grows proportionally to  $\sqrt{Dt}$ . In general, all elevation perturbations will spread proportionally to  $\sqrt{Dt}$  regardless of their initial shape, because the diffusion term of Eq. (1) is linear and thus the superposition principle applies (e.g., Balluffi et al., 2005). Thus, the quantity  $\sqrt{Dt}$  is a length scale that characterizes how far diffusion can spread elevation perturbations during some time *t*. Consequently, the diffusion time scale (e.g., Perron et al., 2008, 2012)

$$t_D \coloneqq \frac{l^2}{D},\tag{35}$$

which is equal to the time needed for the quantity  $\sqrt{Dt}$  to reach some value *l*, is an appropriate measure of diffusion's horizontal influence. Analogously to  $t_l$ , a small  $t_D$  value corresponds to a strong horizontal influence of diffusion.

## 5

15

Following Perron et al. (2008, 2009, 2012), we quantify the relative horizontal influence of incision versus diffusion across a landscape by the ratio of the diffusion time  $t_D$  to the incision time  $t_I$ . Using the definitions in Eqs. (34) and (35) leads to the following definition of the Péclet number Pe:

$$\operatorname{Pe} \coloneqq t_D / t_I = \frac{K \sqrt{A} l}{D} = \frac{\sqrt{A} l}{l_c^2}.$$
(36)

Given that small  $t_D$  or  $t_I$  values correspond to strong influences of the respective process, small Pe values correspond to

10 hillslopes, where diffusion is horizontally dominant, and large Pe values correspond to valleys, where incision is horizontally dominant.

The transition between the regimes of horizontally dominant diffusion and incision can be assumed to occur where the incision and diffusion time scales are roughly equal, i.e., where  $t_I \approx t_D$  or, equivalently, where Pe  $\approx 1$ . Substituting this value into the definition of the Péclet number (Eq. 36) yields

$$Pe \approx 1 \iff \sqrt{A} \ l \approx l_c^2 \ . \tag{37}$$

As mentioned above, Perron et al.'s (2008) Péclet number is equal to one for a length scale equal to  $l_c$ . In contrast, Eq. (37) shows that our Péclet number is equal to one not for specific values of the length scale l or the drainage area A, but rather for all combinations of values for which  $\sqrt{A} l$  is equal to  $l_c^2$ . Note that  $l_c^2 = \sqrt{A_c} l_c$ ; thus, if both  $l \approx l_c$  and  $A \approx A_c$ , then Pe  $\approx$  1. Furthermore, note that for  $l \approx l_c$  and  $A \approx A_c$  the incision and diffusion times are both roughly equal to the

20 characteristic time, i.e.,  $t_I \approx t_c$  and  $t_D \approx t_c$ . Thus, the characteristic length, area, and time scales can be interpreted as characterizing the regime transition from dominant diffusion to dominant incision.

## 4.2.2 Spatial distribution of the Péclet number across a landscape

To calculate values of the Péclet number Pe across a landscape we must specify what the length scale l is. In this study, we define l as the maximum distance along flow paths from a point to the drainage divide, and refer to it as the point's flow path

- length. Because the incision term has the form of a kinematic wave (and kinematic waves propagate only in the direction of the gradient; Lighthill and Whitham, 1955), perturbations are advected only in the uphill direction along flow paths. Thus, it is reasonable to quantify advection's influence using a length scale that is measured along flow paths and points in the uphill direction; the flow path length is such a length scale. Because the diffusion term spreads out elevation differences in all directions (uphill and downhill along flow paths, and laterally across flow paths), its influence could be quantified using
- several length scales, including the flow path length. Because the flow path length is the natural length scale for the incision term, and one of several possible length scales for the diffusion term, it is a reasonable length scale for calculating the Péclet number, which measures the relative influence of advection versus diffusion.

Figure 10 shows how the Péclet number Pe varies across a landscape. We calculated Pe according to Eq. (36), assuming that l is the flow path length and using data from the simulated equilibrium landscape B (as shown in Figs. 1, 2, and 8). Each pixel in Fig. 10 is colored by  $\log_{10}$  Pe. We used the logarithm to visualize values of Pe that are both much smaller and much larger than 1. Figure 10 shows that the highest values of Pe follow the dendritic valley network, similar to the highest values

5 of the incision height  $h_I$  (see Fig. 8). Diffusion and incision are horizontally dominant on hillslopes and in valleys, respectively. Therefore, the dendritic patterns in Fig. 10 suggest that our Péclet number (defined by Eq. (36) and calculated using the flow path length) is a reasonable measure of the relative horizontal influence of incision versus diffusion.

Just as the incision and diffusion heights  $h_I$  and  $h_D$  can be rescaled by the characteristic height  $h_c$  (Eqs. 23 a, b), one can show 10 that, at corresponding points in geometrically similar landscapes, the incision and diffusion times  $t_I$  and  $t_D$  can be rescaled by the characteristic time  $t_c$ , i.e.,

$$t_{I}'/t_{c}' = t_{I}/t_{c}$$
,  $t_{D}'/t_{c}' = t_{D}/t_{c}$ . (38 a, b)

Furthermore, one can show that, at corresponding points in geometrically similar landscapes, Péclet numbers are equal, i.e.,  $Pe' = \sqrt{A'} l' / l_c'^2 = \sqrt{A} l / l_c^2 = Pe$ . (38 c)

In other words, if we would plot maps of Pe using data from the simulated landscapes A and C, then we would obtain rescaled copies of the map of Pe derived from landscape B (as seen in Fig. 10).

## 15 4.2.3 Implications of the new Péclet number definition

In this subsection, we discuss the differences between the Péclet number defined in this study and the Péclet number defined by Perron et al. (2008, 2009, 2012). Our definition (Eq. 36) includes both the drainage area A and the length scale l, whereas theirs included only l. Specifically, our Péclet number is defined as  $Pe = \sqrt{A} l/l_c^2$ , whereas theirs was defined as  $Pe = Kl^2/D = l^2/l_c^2$  (for drainage area and slope exponents m=0.5 and n=1). To introduce their Péclet number, Perron et al.

20 (2008) defined an incision-term celerity that implicitly assumed that  $l = \sqrt{A}$ , and in this way they substituted  $l^2$  for  $\sqrt{A} l$ . In contrast, we defined the incision and diffusion time scales, and the Péclet number (Eqs. 34–36), using an abstract length scale *l*, and we calculated values of the Péclet number assuming that *l* is the flow path length.

To determine how our Péclet number scales with the flow path length *l* or the drainage area *A*, we first need to explore the

- scaling relationship between *l* and *A*. The flow path length *l* scales as a power law of the drainage area *A*, i.e., as  $A^p$ , with an exponent *p* that depends on how convergent or divergent the topography is. (Here, we refer to horizontal topographic convergence or divergence, as measured, for example, by the contour curvature.) In convergent contributing areas, the scaling exponent is p = 0.5 (i.e., *l* scales as  $\sqrt{A}$ ; e.g., Montgomery and Dietrich, 1992), in planar contributing areas, it is p = 1 (i.e., *l* scales linearly with *A*; e.g., Pelletier, 2010), and in divergent topography, it is p > 1. In general, the topography
- 30 of contributing areas is not purely convergent, planar, or divergent; it is a mixture of these three types. In such cases, the scaling exponent *p* will be somewhere between the values of the three types of topography, depending on how they are mixed. At large scales, *l* will generally scale as  $\sqrt{A}$  (e.g., Mueller, 1972), which corresponds to convergent topography.

Figure 11 schematically illustrates how the scaling relationship between flow path length *l* and drainage area *A* depends on the topography of contributing areas. It shows a point inside a valley (P<sub>1</sub>) that has convergent contributing topography, a point on a hillslope (P<sub>2</sub>) that has planar contributing topography, and a point on an interfluve (P<sub>3</sub>) that has divergent contributing topography. Figure 11 shows that, even though the three points have equal flow path lengths, they have different drainage areas, and those drainage areas scale differently with flow path length. Additionally, Fig. 11 illustrates how contributing areas can have mixed topographies; for example, the contributing area of  $P_1$  becomes less convergent and more planar near the ridge.

Given that the Péclet number Pe is proportional to  $\sqrt{A} l$  (Eq. 36) and that l scales as  $A^p$ , the Péclet number will scale as  $l^{1+1/(2p)}$  or  $A^{p+1/2}$ . The values of the exponents 1 + 1/(2p) and p + 1/2 are determined by the value of p, which depends on topography as described above. Therefore, in convergent topography, and at large scales, Pe will scale as  $l^2$  or A. At small scales in non-convergent or mixed topography, Pe will scale with l raised to a power less than 2, or with A raised to a power greater than 1.

Perron et al.'s (2008, 2009, 2012) Péclet number scales as  $l^2$ . To calculate Pe across real-world landscapes, Perron et al. (2012) defined the length scale *l* as the length of basins and calculated it from drainage area data according to  $l = \sqrt{3A}$ ; in this case their Péclet number scales as *A*. Consequently, in convergent topography, and at large scales, both their Péclet

number and ours scale as  $l^2$  or as A. In contrast, at small scales in non-convergent or mixed topography, the two Péclet numbers scale differently from each other with l and A.

#### 5 Summary and conclusions

In this study, we perform dimensional analysis on an LEM that includes terms for stream-power incision, linear diffusion, and uplift (Eq. (1); e.g., Howard, 1994; Dietrich et al., 2003). The governing equation that we analyze in the main text

- (Eq. 1) includes the relatively simple incision term  $K\sqrt{A}|\nabla z|$ , which is a special case of the more general incision law  $KA^m(|\nabla z|)^n$  (Eq. A1). As we demonstrate in Appendix A, results obtained from dimensional analysis of the LEM with the general incision law (Eq. A1) are equivalent to results obtained from the LEM with the simple incision law (Eq. 1), but the latter have much simpler forms and, thus, are more suitable for presentation in the main text.
- Our dimensional analysis is based on two key premises. First, we assume that the dimensions of length and height are conceptually distinct. Second, we use only intrinsic characteristic scales, i.e., scales that depend only on the parameters of the model (the incision coefficient *K*, the diffusion coefficient *D*, and the uplift rate *U*), not on sizes of the domain or of landscape features. We use the characteristic scale  $l_c$  (Eq. 3) previously defined by Perron et al. (2008), and we introduce new characteristic height and time scales  $h_c$  and  $t_c$ . (Eqs. 4 and 5). The use of these three characteristic scales allows us to
- obtain three main results.

First, rescaling the governing equation (Eq. 1) by  $l_c$ ,  $h_c$ , and  $t_c$  yields a dimensionless form (Eq. 16) that includes only variables and no parameters. Because it has no parameters that can be adjusted, this dimensionless equation has only one solution for any given set of (dimensionless) boundary and initial conditions. This result means that landscapes that are

35 rescaled horizontally by  $l_c$  (Eqs. 18 a, b) and whose initial conditions are rescaled vertically by  $h_c$  (Eq. 18 c) will follow temporally and geometrically similar evolutions, i.e., if we compare these landscapes at times that are rescaled by  $t_c$  (Eq. 18 d), then they will be copies of each other (rescaled horizontally by  $l_c$  and vertically by  $h_c$ ). We demonstrate the temporal and geometric similarity of rescaled landscapes theoretically (Appendix B) and numerically (Sect. 3.1, Figs. 1–5).

Second,  $l_c$ ,  $h_c$ , and  $t_c$  can be combined to define other characteristic scales (e.g., characteristic velocities, slopes, and

- curvatures), which rescale variables whose dimensions combine L, H, and T in the same way. Based on these definitions of characteristic scales, we can straightforwardly deduce scaling relations between any landscape metric and the model parameters *K*, *D*, and *U*, because all of our scales are defined to depend only on parameters. As an example, we present a characteristic horizontal velocity that must rescale the migration velocity of drainage divides (Eq. 20).
- The temporal and geometric similarity of rescaled landscapes implies that we can explore all combinations of the model parameters K, D, and U by simulating a single combination of parameters for any given dimensionless domain size, resolution, and initial conditions. We can then simply rescale the results of this simulation to obtain any results for any other combinations of parameters, which is significantly more efficient than running multiple simulations for multiple parameter combinations.

#### 15

Such a modeling approach assumes that simulation domains are rescaled, but this is not always physically realistic (e.g., if a domain represents an island, changing model parameters may change the sizes of ridges and valleys on the island, but not the size of the island itself). Nonetheless, as we explain in Sect. 3.2.2, landscape features that are sufficiently small with respect to the domain size may locally remain statistically similar, even if the landscapes globally are not similar. Therefore, the

20 rescaling approach may offer insights into how such features depend on model parameters, even if the domain is not rescaled.

However, if landscapes are not rescaled and model parameters are varied too widely, then boundary or resolution effects may arise. These boundary and resolution effects will be minimized if the domain size is much larger than  $l_c$  and the resolution is

25 much smaller than  $l_c$ . More generally, we recommend performing model simulations in units of the characteristic scales (i.e., in dimensionless terms) instead of in conventional dimensional units. This should be helpful in comparing disparate model results and identifying which features can be rescaled.

Third,  $l_c$ ,  $h_c$ , and  $t_c$  can be interpreted as expressing the competition between incision, diffusion, and uplift, and as linking the 30 relative strengths of these processes to topographic properties of the landscape (Sects. 4.1, 4.2). This interpretation is facilitated by introducing process-specific height and time scales  $h_l$ ,  $h_D$ ,  $t_l$ , and  $t_D$  (Eqs. 21, 22, 34, and 35) that measure the vertical and horizontal influences of incision and diffusion. The incision and diffusion heights quantify the contribution of these processes to the total elevation change per unit  $t_c$ . The incision and diffusion time scales quantify how long it takes for these processes to propagate elevation perturbations over some horizontal length scale (i.e., the stronger a process, the

35 smaller its time scale).

In steady-state landscapes, the characteristic height  $h_c$  is everywhere equal to the difference between the incision and diffusion height scales, i.e.,  $h_c = h_I - h_D$  (Eq. 29), and in this way  $h_c$  expresses the balance between incision, diffusion, and uplift. Given that we define  $h_I$  as a function of drainage area and slope, and  $h_D$  as a function of curvature,  $h_c = h_I - h_D$ 

expresses how steady-state topography is linked to the balance between incision, diffusion, and uplift (Eq. 30, Figs. 6, 7).

Equations (29) and (30) show that  $h_c$  expresses the balance between all three processes, not only between incision and uplift as the definition  $h_c = U/K$  (Eq. 4) seemingly suggests. This can also be seen by the fact that, for the case of the LEM with the general incision law (Eq. A1), the definition of  $h_c$  includes all three parameters (Eq. A9).

Equation (30) is a testable prediction that can discriminate between landscapes that are in steady state and follow the governing equation (Eq. 1), and those that do not. Furthermore, it can be rearranged to estimate  $l_c$  and  $h_c$ , i.e., the parameter ratios D/K and U/K. Finally, it implies that the incision height  $h_l$  is linearly related to curvature (Eq. 33). Thus, the spatial distribution of  $h_l$  follows the valley network (Fig. 8), and  $h_l$  can be used as a proxy for curvature to define the hillslope–valley transition.

The characteristic length and time scales  $l_c$  and  $t_c$  characterize the regime transition from diffusion-dominated to incisiondominated propagation of topographic perturbations. Following Perron et al. (2008, 2009, 2012), we quantify the relative influence of incision versus diffusion by the ratio of the diffusion timescale  $t_D$  to the incision time scale  $t_I$ . This leads to a new definition of the Péclet number (specifically,  $Pe = \sqrt{A} l/l_c^2$ , where *l* is the flow path length to the divide; Eq. 36). The

15 spatial distribution of this Péclet number follows the valley network (Fig. 10), with small Pe values corresponding to diffusion-dominated features (ridges and hillslopes) and large Pe values corresponding to incision-dominated features (valleys). The transition occurs where Pe is roughly equal to one (i.e., where  $t_l$  and  $t_D$  are roughly equal). This condition is satisfied by combinations of length scale *l* and drainage area *A* for which  $\sqrt{A} \ l \approx l_c^2$  (Eq. 37). One such combination is  $l \approx l_c$ and  $A \approx l_c^2$ , in which case  $t_l$  and  $t_D$  are both also roughly equal to  $t_c$ .

Our definition of the Péclet number differs from Perron et al.'s (2008, 2009, 2012) in that ours includes both the length scale l and the drainage area A, while theirs includes only l. Perron et al.'s definition implicitly assumes that  $l = \sqrt{A}$ . We assume that l is the flow path length, i.e., the maximum distance along the flow paths from a point to the divide. In this way our Pe expresses how topographic convergence and divergence control the relative importance of incision versus diffusion across

the landscape (Sect. 4.2.3, Fig. 11). In convergent topography, the two Pe definitions (ours and Perron et al.'s) both scale as  $l^2$ . However, in planar or divergent topographies, Perron et al.'s Pe still scales as  $l^2$  whereas ours scales as  $l^p$  with  $p \le 3/2$ .

To summarize,  $l_c$ ,  $h_c$ , and  $t_c$  lead to a dimensionless form of the governing equation that significantly simplifies the model. Additionally, they can be combined to rescale any variables with dimensions in L, H, and T. Finally, they express process

- competitions and link them to topographic properties. The ability of  $l_c$ ,  $h_c$ , and  $t_c$  to perform all of these tasks is a direct consequence of the two key premises underlying our dimensional analysis (characteristic scales are intrinsic, and horizontal lengths and vertical heights are dimensionally distinct). Our analysis suggests that  $l_c$ ,  $h_c$ , and  $t_c$ , as a group, are fundamental properties of landscapes that follow equations such as Eqs. (1) or (A1). This in turn suggests that  $l_c$ ,  $h_c$ , and  $t_c$ , or their combinations, may explain additional properties of ridge and valley topography, beyond those that we briefly mention here
- for demonstration purposes. Therefore, it may be illuminating to estimate  $l_c$ ,  $h_c$ , and  $t_c$  in future landscape studies, whether in the field, in laboratory experiments, or in computer simulations.

## A1 Generic governing equation

We perform dimensional analysis of an LEM with an incision term with generic drainage area and slope exponents m and n that follows the governing equation

$$\frac{\partial z}{\partial t} = -KA^m (|\nabla z|)^n + D\nabla^2 z + U.$$
(A1)

#### 5 A2 Dimensions and characteristic scales

The variables and parameters of Eq. (A1) have the following dimensions. Coordinates of points (x, y) have dimensions of L, elevation z has dimensions of H, and time t has dimensions of T. Therefore, using square brackets to denote the dimensions of quantities, we obtain  $[\partial z/\partial t] = H T^{-1}$ ,  $[A] = L^2$ ,  $[|\nabla z|] = H L^{-1}$ ,  $[\nabla^2 z] = H L^{-2}$ . Consequently, the coefficient of incision K has dimensions of

$$[K] = \frac{[\partial z/\partial t]}{[A^m(|\nabla z|)^n]} = \frac{\mathrm{H}\,\mathrm{T}^{-1}}{\mathrm{L}^{2m}\,\mathrm{H}^n\,\mathrm{L}^{-n}} = \mathrm{L}^{n-2m}\,\mathrm{H}^{1-n}\,\mathrm{T}^{-1}\,.$$
(A2)

Likewise, the coefficient of diffusion and the uplift rate have dimensions of

$$[D] = L^2 T^{-1}, (A3)$$

$$[U] = H T^{-1}.$$
(A4)

All terms of Eq. (A1) have dimensions that can be expressed in terms of L, H, and T. Therefore, we can non-dimensionalize Eq. (A1) using a characteristic length  $l_c$ , a characteristic height  $h_c$ , a characteristic time  $t_c$ , and combinations thereof. We wish to define  $l_c$ ,  $h_c$ , and  $t_c$  as functions of only the parameters K, D, and U, and the exponents m and n. We can utilize Eqs. (A2)–

15 (A4) to derive these functions. Below, we present how the definition of  $l_c$  is obtained. The definitions of  $h_c$  and  $t_c$  can be obtained analogously.

We assume that  $l_c$  will be the product of three power laws of the three parameters K, D, and U, i.e.,  $l_c = K^k D^d U^u$ , where k, d, and u are unknown exponents that we seek to determine. The two sides of this definition of  $l_c$  must have the same dimensions i.e.  $[L] = [K^k D^d U^k]$  which implies in combination with Eqs. (A2) (A4) that

dimensions, i.e.,
$$[l_c] = [K^k D^d U^u]$$
, which implies, in combination with Eqs. (A2)–(A4), that  

$$L = [K]^k [D]^d [U]^u = (L^{n-2m} H^{1-n} T^{-1})^k (L^2 T^{-1})^d (H T^{-1})^u = L^{(n-2m)k+2d} H^{(1-n)k+u} T^{-k-d-u}.$$
(A5)

We can find the exponents k, d, and u by requiring that in the right-hand side of Eq. (A5), the exponent of L is equal to one and the exponents of H and T are equal to zero, i.e., by solving the system

$$\begin{cases} (n-2m)k +2d = 1\\ (1-n)k +u = 0.\\ -k -d -u = 0 \end{cases}$$
The solution of this system is
$$k = \frac{-1}{n+2m}, \quad d = \frac{n}{n+2m}, \quad u = \frac{1-n}{n+2m},$$
(A7)

i.e.,
$$L = ([K]^{-1}[D]^n[U]^{1-n})^{1/(n+2m)}$$

Therefore, the characteristic length can be defined as:

$$l_{c} = (K^{-1}D^{n}U^{1-n})^{1/(n+2m)}.$$
(A8)  
Following the same procedure we find definitions of the characteristic height and time scales:  

$$h_{c} = (K^{-2}D^{n-2m}U^{2-n+2m})^{1/(n+2m)},$$
(A9)

$$t_c = (K^{-2}D^{n-2m}U^{2-2n})^{1/(n+2m)}.$$
(A10)

For m = 0.5 and n = 1, Eqs. (A8)–(A10) yield the corresponding Eqs. (3)–(5) in the main text.

5

Note the exponents of the parameters *K*, *D*, and *U* in the definitions of  $l_c$ ,  $h_c$ , and  $t_c$  (Eqs. A8–A10). The following observations can be made:

- For n = 1, the definitions of  $l_c$  and  $t_c$  are simplified, because U drops out.
- For n = 2m, the definitions of  $h_c$  and  $t_c$  are simplified, because D drops out.
- *D* would drop out of  $l_c$ 's definition if n = 0 and *U* would drop out of  $h_c$ 's definition if n = 2 + 2m. Such exponents would not yield physically meaningful stream-power or shear-stress incision laws.
  - *K* never drops out of any of the three definitions.
  - Therefore, eliminating U from  $l_c$ 's and  $t_c$ 's definitions (for n = 1), or eliminating D from  $h_c$ 's and  $t_c$ 's definitions (for n = 2m) are the only simplifications of the definitions of characteristic scales that correspond to meaningful stream-
- power or shear-stress incision laws.
  - Equation (1) satisfies both n = 1 and n = 2m; therefore, it results in the simplest set of characteristic scales that are physically meaningful.

Based on the definitions of 
$$l_c$$
,  $h_c$ , and  $t_c$  we define a characteristic area

$$A_{c} = l_{c}^{2} = (K^{-2}D^{2n}U^{2-2n})^{1/(n+2m)},$$
(A11)

a characteristic gradient

$$G_c = h_c / l_c = (K^{-1} D^{-2m} U^{1+2m})^{1/(n+2m)},$$
(A12)

and a characteristic curvature

$$\kappa_c = G_c / l_c = h_c / l_c^2 = U / D .$$
(A13)

(A14)

Note that the uplift rate can be viewed as a characteristic rate of elevation change, because  $h_c/t_c = U$ .

We define a characteristic horizontal velocity as

$$u_c = l_c / t_c = (K^1 D^{2m} U^{-1+n})^{1/(n+2m)}.$$
(A15)

We have shown that the characteristic scales of the governing equation in the main text, Eq. (1), are consistent with the characteristic scales of Eq. (A1). Below we confirm that all results and interpretations that refer to Eq. (1) and were presented in the main text are also consistent with Eq. (A1).

#### A3 Dimensionless governing equation

We define dimensionless variables according to Eqs. (6, 8, 10, 12, and 14) (using the characteristic scales defined by Eqs. A8–A13) and substitute into Eq. (A1):

$$\frac{\partial z}{\partial t} = -KA^{m}(|\nabla z|)^{n} + D\nabla^{2}z + U \Leftrightarrow 
\frac{h_{c}\partial z^{*}}{l_{c}\partial t^{*}} = -KA_{c}^{m}A^{*m}G_{c}^{n}(|\nabla^{*}z^{*}|)^{n} + D\kappa_{c}\nabla^{*2}z^{*} + U \Leftrightarrow ... \Leftrightarrow U\frac{\partial z^{*}}{\partial t^{*}} = U(-A^{*m}(|\nabla^{*}z^{*}|)^{n} + \nabla^{*2}z^{*} + 1) \Leftrightarrow 
\frac{\partial z^{*}}{\partial t^{*}} = -A^{*m}(|\nabla^{*}z^{*}|)^{n} + \nabla^{*2}z^{*} + 1.$$
(A16)

## A4 Height scales

5 We define an incision height as the erosion due to incision during one unit of  $t_c$ 

 $h_{I} = KA^{m}(|\nabla z|)^{n} t_{c} = A^{m}(|\nabla z|)^{n} (K^{n+2m-2}D^{n-2m}U^{2-2n})^{1/(n+2m)}.$ (A17)

Note that the dimensionless incision height is:

$$\frac{h_I}{h_c} = \frac{KA^m (|\nabla z|)^n t_c}{h_c} = \frac{A^m (|\nabla z|)^n}{U/K}.$$
(A18)

The incision height  $h_l$  is related with the steepness index  $k_s$  according to:

$$k_s = A^{m/n} |\nabla z| = \left(\frac{h_I}{K t_c}\right)^{1/n}.$$
(A19)

10 We define a diffusion height as the elevation change due to diffusion per unit of  $t_c$ 

 $h_D = D\nabla^2 z \, t_c = (D \, t_c) \, \nabla^2 z = (K^{-1} D^n U^{1-n})^{2/(n+2m)} \, \nabla^2 z = l_c^2 \, \nabla^2 z \,. \tag{A20}$ 

Note that the dimensionless diffusion height is:

$$\frac{h_D}{h_c} = \frac{D\nabla^2 z t_c}{h_c} = \frac{\nabla^2 z}{U/D} = \frac{\nabla^2 z}{\kappa_c}.$$
(A21)

Multiplying Eq. (A1) by  $t_c$  and substituting from Eqs. (A14), (A17), and (A20) we obtain

$$(\partial z/\partial t)t_c = -KA^m (|\nabla z|)^n t_c + D\nabla^2 z t_c + U t_c \quad \Leftrightarrow \quad (\partial z/\partial t)t_c = -h_I + h_D + h_c ,$$
(A22)  
which in steady state becomes

$$0 = -h_I + h_D + h_c \,. \tag{A23}$$

At points of zero curvature ( $\nabla^2 z = 0$ ), Eq. (A23) becomes

$$h_{I} = h_{c} \iff A^{m}(|\nabla z|)^{n} (K^{n+2m-2}D^{n-2m}U^{2-2n})^{1/(n+2m)} = (K^{-2}D^{n-2m}U^{2-n+2m})^{1/(n+2m)} \iff A^{m}(|\nabla z|)^{n} = U/K.$$
(A24)

Therefore,  $h_c$  and U/K are the steady state values of  $h_l$  and  $A^m(|\nabla z|)^n$ , respectively, at points of zero curvature. For m = 0.5 and n = 1, i.e., for the governing equation in the main text (Eq. 1), these two relations coincide.

At drainage divides 
$$(A = 0)$$
, Eq. (A23) becomes  
 $h_D = -h_c \iff D\nabla^2 z \ t_c = -U \ t_c \iff \nabla^2 z = -U/D = -\kappa_c .$  (A25)

Substituting the definition of  $h_D$  (Eq. A20) into Eq. (A23) yields:

$$h_I = l_c^2 \,\nabla^2 z + h_c \,. \tag{A26}$$

i.e.,  $h_I$  remains a linear function of curvature in steady state. Thus,  $h_I$  can be used for valley definition for the generic model 5 in the same way that it can be used for the simplified model.

## A5 Length and time scales

The celerity of the incision term is:

$$c(A, |\nabla z|) = KA^m (|\nabla z|)^{n-1}.$$
(A27)

Therefore, the time needed to advect perturbations over some distance l defines an incision time scale:

$$t_I = \frac{\iota}{KA^m(|\nabla z|)^{n-1}}.$$
(A28)

The diffusion time scale of Eq. (A1) is the same as in the case of Eq. (1), i.e. according to Eq. (35), it is  $t_D = l^2/D$ .

Therefore, the Péclet number is defined as

$$Pe = t_D / t_I = \frac{A^m (|\nabla z|)^{n-1} l}{D/K}.$$
(A29)

We observe that  $l_c^{2m+1} G_c^{n-1} = D/K$ . Therefore, the Péclet number definition becomes:

$$\operatorname{Pe} = \frac{A^m (|\nabla z|)^{n-1} l}{l_c^{2m+1} G_c^{n-1}} = \left(\frac{A}{A_c}\right)^m \left(\frac{|\nabla z|}{G_c}\right)^{n-1} \left(\frac{l}{l_c}\right).$$
(A30)

The condition  $Pe \approx 1$  is satisfied at points with length scale *l*, drainage area *A*, and slope  $|\nabla z|$  such that:

$$\operatorname{Pe} \approx 1 \iff \left(\frac{A}{A_c}\right)^m \left(\frac{|\nabla z|}{G_c}\right)^{n-1} \left(\frac{l}{l_c}\right) \approx 1 \iff A^m |\nabla z|^{n-1} l \approx l_c^{2m+1} G_c^{n-1}.$$
(A31)

One combination of length scales, drainage areas, and slopes that satisfies this condition is  $l \approx l_c$ ,  $A \approx A_c$ , and  $|\nabla z| \approx G_c$ . Substituting these values into the definitions of  $t_l$  and  $t_D$  yields  $t_l \approx t_c$  and  $t_D \approx t_c$ .

## **Appendix B: Rescaling of landscapes**

We outline an analytical proof of the rescaling property implied by the dimensionless governing equation (Eq. 16), namely that landscapes with any parameters will evolve temporally and geometrically similarly and will reach geometrically similar steady states provided that we properly rescale their boundary and initial conditions. Equations (18 a–d) defines what we term as proper rescaling.

A simple way to demonstrate this property is to explore the necessary conditions for the dimensionless governing equation (Eq. 16) to describe two different landscapes and then show that the same conditions lead to temporally and geometrically similar evolution of these two landscapes. Let a landscape have parameters K, D, and U and a second landscape have parameters K' = kK, D' = dD, and U' = uU, where k, d, and u are positive real numbers, and let both landscapes satisfy Eq. (16). In what follows, we denote variables and scales of the second landscape as primed to match the notation of its

Eq. (16). In what follows, we denote variables and scales of the second landscape as primed to match the notation of its parameters.

First, we derive rescaling relationships between characteristic scales of the two landscapes as functions of k, d, and u. The second landscape's characteristic length scale,  $l_c'$ , will be

$$l_c' = \sqrt{D'/K'} = \sqrt{dD/kK} = \sqrt{d/k} l_c.$$
(B1 a)

10 Likewise, its remaining characteristic scales will be

$$h_c' = (u/k) h_c$$
,  $t_c' = (1/k) t_c$ ,  $A_c' = (d/k) A_c$ ,  $G_c' = (u/\sqrt{dk}) G_c$ ,  $\kappa_c' = (u/d) \kappa_c$ . (B1 b-f)

Second, we derive relationships between coordinates of the landscapes and between moments in time during their evolution. We substitute Eq. (B1 a) into Eq. (18 a) and obtain

$$x' = (l_c'/l_c) x = \sqrt{d/k} x.$$
 (B2 a)

Likewise, we can obtain

$$y' = \sqrt{d/k} y, \tag{B2 b}$$

t' = (1/k) t. (B2 c)

We consider the two landscapes to be temporally and geometrically similar if the points (x, y) and (x', y') have elevations z and z' at moments in time t and t', respectively, such that

$$z'(t') = (u/k) z(t)$$
. (B2 d)

Third, one can show that drainage areas, time derivatives, and differential operators of the two landscapes will be related according to

$$A' = (d/k) A, \quad \partial/\partial t' = k \quad \partial/\partial t, \quad \nabla' = \sqrt{k/d} \quad \nabla, \quad {\nabla'}^2 = (k/d) \quad \nabla^2.$$
(B2 e-h)

Fourth, we can retrieve the following dimensional governing equation for the second landscape if we start from the dimensionless Eq. (16) and use the rescaling formulas of Eq. (18):

$$\frac{\partial z'}{\partial t'} = -K'\sqrt{A'}|\nabla'z'| + D'\nabla'^2z' + U'.$$
(B3)

We can retrieve Eq. (1), the first landscape's dimensional governing equation, by reversing the derivation of Eq. (16)

presented in Sect. 2.2. In Eq. (B3) it is important to note that we use primed elevations, time, drainage areas, and differential operators. This is the necessary condition for the dimensionless Eq. (16) to describe both landscapes, and we found it by using Eqs. (B2 a–h) to rescale variables between the two landscapes.

Fifth, we can now show that rescaling according to Eqs. (B2 a–h) additionally leads to temporal and geometric similarity. 30 Specifically, we can show that if the two landscapes are geometrically similar (i.e., they obey Eqs. B2 d) at any moments in time *t* and *t'* that are rescaled according to Eq. (B2 c), then they will remain geometrically similar for all later moments in time. To show this we compare the left-hand side and the incision, diffusion, and uplift terms of the right-hand side of the dimensional governing equations of the two landscapes. These terms have dimensions of H T<sup>-1</sup>, i.e., dimensions of a vertical velocity. Thus, to simplify notation, we denote them here as v,  $v_I$ ,  $v_D$ , and  $v_U$ , respectively. The comparisons show that

$$v_I'(t') = -K'\sqrt{A'}|\nabla' z'(t')| = -kK\sqrt{d/k}A\sqrt{k/d}|\nabla((u/k)z(t))| = -uK\sqrt{A}|\nabla z(t)| = uv_I(t), \quad (B4 a)$$
  
likewise,

$$v_D'(t') = u v_D(t), \quad v_U'(t') = u v_U(t),$$
 (B4 b-c)

and, therefore, also

$$v'(t') = v_I'(t') + v_D'(t') + v_U'(t') = u \left( v_I(t) + v_D(t) + v_U(t) \right) = u v(t).$$
(B4 d)

Thus, after a time interval dt' the second landscape will have elevation equal to

z'(t' + dt') = z'(t') + v'(t')dt' = (u/k)z(t) + u v(t) (1/k)dt = (u/k)(z(t) + v(t)dt) = (u/k)z(t + dt), (B5) i.e., the two landscapes will continue obeying Eq. (B2 d) and thus remain geometrically similar.

Therefore, we can conclude that if the two landscapes have geometrically similar initial conditions, then they will evolve temporally and geometrically similarly and will reach geometrically similar steady states.

# Appendix C: Numerical simulation set-up

## C1 Inputs

#### C1.1 Calculation of CHILD parameters

- To simulate the governing equation (Eq. 1) with CHILD we used the detachment-limited module, constant, uniform, and continuous precipitation, zero infiltration, hydraulic geometry scaling exponents  $\omega_b$  and  $\omega_s$  equal to 0.5, and detachment capacity exponents  $m_b$ ,  $n_b$ , and  $P_b$  equal to 1 (see Tucker et al., 2001, and Tucker, 2010, for definitions of CHILD's assumptions, modules, and parameters).
- For this choice of exponents, the rate of elevation change due to incision is calculated by CHILD from the equations (in CHILD notation)

$$\frac{\partial z}{\partial t}\Big|_{\text{Incision}} = -D_c = -k_b \tau , \qquad (C1 \text{ a})$$
$$\tau = k_t \frac{\sqrt{P}\sqrt{A}}{k_w} S , \qquad (C1 \text{ b})$$

where  $D_c$  is the maximum detachment capacity in ma<sup>-1</sup>,  $\tau$  is stream power per unit bed area in Wm<sup>-2</sup>,  $k_b$  is a detachment rate coefficient in m a<sup>-1</sup> (W m<sup>-2</sup>)<sup>-1</sup> (i.e.,  $k_b$  is the rate of elevation change per each unit of stream power per unit bed area),  $k_t$  is the specific weight of water in N m<sup>-3</sup>, P is the precipitation intensity in m a<sup>-1</sup>,  $k_w$  is bankfull width per unit scaled discharge in

 $25 ext{ s}^{0.5} ext{ m}^{-0.5}$ , and *S* is slope (Tucker et al., 2001; Tucker, 2010).

Equating the incision term of Eq. (1) to  $D_c$ , we can relate the incision coefficient, K, with CHILD's parameters according to

$$K = \frac{k_b k_t \sqrt{P}}{k_w} \frac{\sqrt{1 a}}{\sqrt{31557600 s}}.$$
 (C2)

CHILD input files use a mixed system of units which includes both years and seconds and the program converts variables to a single system of units. Therefore, to calculate the values of parameters entered into CHILD's input files, we must include the unit conversion factor seen in Eq. (C2).

5

We varied the values of K by varying the value of  $k_b$  according to Eq. (C2), while we used constant values of  $k_t = 9810 \text{ N m}^{-3}$ ,  $P \approx 1.31 \text{ m a}^{-1}$ , and  $k_w = 10 \text{ s}^{0.5} \text{m}^{-0.5}$ .

We defined combinations of the parameters *K*, *D*, and *U*, each ranging by two orders of magnitude around arbitrary baseline
values. The baseline values were found to be used frequently in the literature (e.g., Perron et al., 2008; Tucker, 2009; Clubb et al., 2016). Table 2 shows values of parameters, characteristic scales, and simulation properties of the three landscapes presented in Figs. 1–5 and 7–10, including the baseline landscape (landscape B).

## C1.2 Simulation domain and mesh

We synthesized two random TINs by randomly perturbing a deterministic TIN generated by the geometry definition module
 of MATLAB's PDE toolbox. We used Matlab to better control the rescaling procedure that we describe below. We exported the rescaled TINs of simulations as text files. Using the coordinates of TIN points included in these files, CHILD calculated the corresponding Delaunay triangulation using its own, built-in modules. We set all four domain boundaries to be open (see Tucker, 2010).

- To prepare rescaled TINs, we followed the following procedure. First, we synthesized two random TINs on rectangular domains with *x*-coordinates between 0 and 200 and *y*-coordinates between 0 and 400; the average triangle edge length was 0.8, and the *z*-coordinates were drawn from a uniform distribution between 0 and 0.1. By assuming that the aforementioned values are unitless, we can consider the resulting TINs to be dimensionless. Second, using Eqs. (3) and (4), we calculated the characteristic length and height scales  $l_c$  and  $h_c$  that correspond to the parameter combination of each simulation. Third, using
- Eq. (6), we multiplied the *x*-, *y*-, and *z*-coordinates of the points of the dimensionless TINs by the  $l_c$  and  $h_c$  of each simulation. This yielded each simulation's dimensional TINs with size of 200  $l_c$  by 400  $l_c$ , an average triangle edge of 0.8  $l_c$ , and initial elevations between 0 and 0.1  $h_c$ . CHILD assumes that *x*, *y*, and *z* are in units of meters; therefore, we calculated  $l_c$  and  $h_c$  in units of meters as well. The above procedure resulted in dimensional TINs that are rescaled copies of each other according to Eqs. (18 a–c).

## 30 C1.3 Time steps

We defined the time step length  $\Delta t$  using Courant-Friedrichs-Lewy criteria for the incision and diffusion terms of Eq. (1) according to the formulas (Refice et al, 2012):

 $\Delta t = \min[\Delta t_{Inc.}, \Delta t_{Diff.}],$ with
(C3 a)

$$\Delta t_{Inc.} = \min\left[\frac{L_{ij}}{K\sqrt{A_i}}\right],\tag{C3 b}$$

$$\begin{bmatrix} L_{ij}^2 \end{bmatrix}$$

$$\Delta t_{Diff.} = \min\left[\frac{L_{ij}}{2D}\right],\tag{C3 c}$$

where  $L_{ij}$  is the length of the TIN edge connecting points *i* and *j* and  $A_i$  is the drainage area of point *i*. Normalizing lengths and drainage areas in Eqs. (C3 b, c) by  $l_c$  and  $l_c^2$ , respectively, yields

$$\Delta t_{Inc.} = \min\left[\frac{l_c L_{ij}^*}{K\sqrt{l_c^2 A_i^*}}\right] = \frac{1}{K} \min\left[\frac{L_{ij}^*}{\sqrt{A_i^*}}\right] = t_c \min\left[\frac{L_{ij}^*}{\sqrt{A_i^*}}\right] = t_c \Delta t_{Inc.}^*,$$
(C4 a)

$$\Delta t_{Diff.} = \min\left[\frac{l_c^2 L_{ij}^{*2}}{2D}\right] = \frac{1}{K} \min\left[\frac{L_{ij}^{*2}}{2}\right] = t_c \min\left[\frac{L_{ij}^{*2}}{2}\right] = t_c \Delta t_{Diff.}^*$$
(C4 b)

Therefore, both the incision and diffusion time steps limits are in effect rescaled by  $t_c$ , even though we do not explicitly impose such scaling.

5

CHILD allows the definition of a single time step length, which is used throughout the entire simulation. However, since  $A_i$  evolves during simulations we do not know a priori which combination of  $L_{ij}$  and  $A_i$  will yield the smallest  $\Delta t_{lnc.}$ . Therefore, to err on the safe side, we calculate  $\Delta t_{lnc.}$  using the domain-wide minimum of  $L_{ij}$  and an  $A_i$  equal to  $0.5(100l_c)^2$ , which is the area of a square whose diagonal spans the domain's half width. We assume that this square has a similar area as the largest

basins to be formed. To avoid very small  $L_{ij}$  which would result in very short  $\Delta t$ , we do not allow TIN perturbation during the synthesis of random TINs (described above) to result in  $L_{ij}$  shorter than 1/3 of the average TIN length of  $0.8l_c$ .

#### C1.4 Steady state

We assume that  $\varepsilon = 0.001$  in Eq. (19), i.e., we assume that a steady state has been reached when

$$\max\left[\frac{|\Delta z_i|}{\Delta t}\right] \le 0.001U,$$
(C5)

where  $\Delta z_i$  is the elevation change of point *i* during a time step  $\Delta t$ .

## 15 C2 Outputs

CHILD produces output files with various variables. Relevant to our model are those with data of elevation,  $z_i$ , drainage area,  $A_i$ , slope,  $|\nabla z_i|$ , and stream power per unit stream bed area,  $\tau_i$ , at every point *i*, the point IDs of TIN triangle and edge vertices, and the ID of the neighbor to which each point drains.

20 Using the triangle and edge output data, we define the Voronoi polygon associated with a point *i*, and calculate this point's curvature  $\nabla^2 z_i$  and elevation change rate due to diffusion  $v_{D_i}$  according to the formulas (Tucker et al, 2001)

$$\nabla^2 z_i = -\frac{1}{\Lambda_i} \sum_{j=1}^{M_i} \frac{z_i - z_j}{\lambda_{ij}} w_{ij},$$

$$v_{D_i} = D \nabla^2 z_i,$$
(C6)
(C7)

where  $\Lambda_i$  is the area of the Voronoi polygon of point *i*, *j* are neighbors of *i*,  $M_i$  is the number of these neighbors,  $z_i$  and  $z_j$  are elevations of points *i* and *j*,  $\lambda_{ij}$  is the distance between *i* and *j*, and  $w_{ij}$  is the length of the Voronoi polygon edge shared between *i* and *j*.

## Author contribution

NT derived analytical results, and NT and JWK interpreted them. NT designed, performed, and analyzed numerical simulations, and NT, HS, and JWK interpreted them. NT drafted the manuscript, and NT, HS, and JWK edited it.

#### **Competing interests**

The authors declare that they have no conflict of interest.

#### Acknowledgments

This study was made possible by financial support from ETH Zurich. We thank Taylor Perron and Sean Willett for helpful discussions, and Greg Tucker and an anonymous referee for their feedback, which helped improve our manuscript.

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

#### 25 doi:10.1002/esp.3302, 2013.

Perron, J. T., Dietrich, W. E., and Kirchner, J. W.: Controls on the spacing of first-order valleys, J. Geophys. Res., 113(F4), doi:10.1029/2007JF000977, 2008.

Perron, J. T., Richardson, P. W., Ferrier, K. L., and Lapôtre, M.: The root of branching river networks, Nature, 492(7427), 100–103, doi:10.1038/nature11672, 2012.

Refice, A., Giachetta, E., and Capolongo, D.: SIGNUM: A Matlab, TIN-based landscape evolution model, Comput. Geosci., 45, 293–303, doi:10.1016/j.cageo.2011.11.013, 2012.

Robl, J., Hergarten, S., and Prasicek, G.: The topographic state of fluvially conditioned mountain ranges, Earth-Sci. Rev., 168, 190-217,

doi:10.1016/j.earscirev.2017.03.007, 2017.

Roering, J. J., Kirchner, J. W., and Dietrich, W. E.: Evidence for nonlinear, diffusive sediment transport on hillslopes and implications for landscape morphology, Water Resour. Res., 35(3), 853–870, doi:10.1029/1998WR900090, 1999.

Roering, J. J., Perron, J. T., and Kirchner, J. W.: Functional relationships between denudation and hillslope form and relief, Earth Planet. Sci. Lett., 264(1–2), 245–258, doi:10.1016/j.epsl.2007.09.035, 2007.

Simpson, G. and Schlunegger, F.: Topographic evolution and morphology of surfaces evolving in response to coupled fluvial and hillslope sediment transport, J. Geophys. Res. Solid Earth, 108(B6), doi:10.1029/2002JB002162, 2003.

Sklar, L. and Dietrich, W. E.: River longitudinal profiles and bedrock incision models: Stream power and the influence of sediment supply, in: Rivers Over Rock: Fluvial Processes in Bedrock Channels, Geophys. Monogr. Ser., Vol. 107, edited by Tinkler, K. and Wohl, E., 237–260, AGU, Washington, D. C., USA, 1998.

Perron, J. T., Kirchner, J. W., and Dietrich, W. E.: Formation of evenly spaced ridges and valleys, Nature, 460(7254), 502–505, doi:10.1038/nature08174, 2009.

Smith, T. R. and Bretherton, F. P.: Stability and the conservation of mass in drainage basin evolution, Water Resour. Res., 8(6), 1506–1529, 1972.

Sonin, A. A.: The Physical Basis of Dimensional Analysis, 2<sup>nd</sup> edition, Department of Mechanical Engineering, MIT, Cambridge, MA, USA, available at: http://web.mit.edu/2.25/www/pdf/DA\_unified.pdf, last access: 30 March 2018, 2001.

Tucker, G. E.: Natural experiments in landscape evolution, Earth Surf. Process. Landf., 34(10), 1450–1460, doi:10.1002/esp.1833, 2009. Tucker, G. E.: CHILD Users Guide for version R9.4.1, Coop. Inst. Res. Environ. Sci. (CIRES) and Dep. Geol. Sci., Univ. Colo., Boulder, CO, USA, available at: http://csdms.colorado.edu/mediawiki/images/Child\_users\_guide.pdf, last access: 30 March 2018, 2010. Tucker, G. E. and Hancock, G. R.: Modelling landscape evolution, Earth Surf. Process. Landf., 35(1), 28–50, doi:10.1002/esp.1952, 2010. Tucker, G., Lancaster, S., Gasparini, N., and Bras, R.: The channel-hillslope integrated landscape development model (CHILD), in:

Landscape erosion and evolution modeling, edited by Harmon, R.S. and Doe, W.W., III, 349–388, Kluwer Academic/Plenum Publishers, New York, USA, 2001.

Whipple, K. X.: Fluvial landscape response time: How plausible is steady-state denudation?, Am. J. Sci., 301(4–5), 313–325, 2001.
Whipple, K. X. and Tucker, G. E.: Dynamics of the stream-power river incision model: Implications for height limits of mountain ranges, landscape response timescales, and research needs, J. Geophys. Res. Solid Earth, 104(B8), 17661–17674, doi:10.1029/1999JB900120,

1999.

Whipple, K. X., Forte, A. M., DiBiase, R. A., Gasparini, N. M., and Ouimet, W. B.: Timescales of landscape response to divide migration and drainage capture: Implications for the role of divide mobility in landscape evolution, J. Geophys. Res. Earth Surf., 122(1), 248–273, doi:10.1002/2016JF003973, 2017.

Willett, S. D., McCoy, S. W., Perron, J. T., Goren, L., and Chen, C.-Y.: Dynamic Reorganization of River Basins, Science, 343(6175),

1248765–1248765, doi:10.1126/science.1248765, 2014.

Willgoose, G., Bras, R. L., and Rodriguez-Iturbe, I.: A coupled channel network growth and hillslope evolution model: 2. Nondimensionalization and applications, Water Resour. Res., 27(7), 1685–1696, doi:10.1029/91WR00936, 1991.

Wobus, C., Whipple, K. X., Kirby, E., Snyder, N., Johnson, J., Spyropolou, K., Crosby, B., and Sheehan, D.: Tectonics from topography: Procedures, promise, and pitfalls, in: Tectonics, Climate, and Landscape Evolution: Geological Society of America Special Paper 398,

Penrose Conference Series, edited by Willett, S.D., Hovius, N., Brandon, M.T., and Fisher, D.M., Geological Society of America, Boulder, CO, USA, 55–74, doi: 10.1130/2006.2398(04), 2006

# Tables

Table 1: List of symbols (in the order they appear)

| Symbol       | Dimensions   |                        | Definition or first use |
|--------------|--------------|------------------------|-------------------------|
|              | (L: length,  | Description            |                         |
|              | H: height,   | Description            |                         |
|              | T: time)     |                        |                         |
| (x, y)       | L            | Horizontal coordinates | Eq. (1)                 |
| Z            | Н            | Elevation              | Eq. (1)                 |
| t            | Т            | Time                   | Eq. (1)                 |
| K            | $T^{-1}$     | Incision coefficient   | Eq. (1)                 |
| D            | $L^2 T^{-1}$ | Diffusion coefficient  | Eq. (1)                 |
| U            | $H T^{-1}$   | Uplift rate            | Eq. (1)                 |
| Α            | $L^2$        | Drainage area          | Eq. (1)                 |
| $ \nabla z $ | $H L^{-1}$   | Topographic slope      | Eq. (1)                 |

| $\nabla^2 z$                      | $H L^{-2}$        | Curvature                                       | Eq. (1)                              |  |
|-----------------------------------|-------------------|-------------------------------------------------|--------------------------------------|--|
| m                                 | [-]               | Drainage area exponent                          | Eq. (A1)                             |  |
| n                                 | [-]               | Slop exponent                                   | Eq. (A1)                             |  |
| З                                 | $H T^{-1}$        | Steady-state threshold                          | Eq. (2)                              |  |
| $l_c$                             | L                 | Characteristic length                           | Eq. (3)                              |  |
| $h_c$                             | Н                 | Characteristic height                           | Eq. (4)                              |  |
| $t_c$                             | Т                 | Characteristic time                             | Eq. (5)                              |  |
| $(x^*, y^*), z^*, \nabla^*,$ etc. | [-]               | Dimensionless variables, operators, etc.        | Eq. (16)                             |  |
| i, j                              | L                 | Unit vectors                                    | Eq. (9)                              |  |
| $G_c$                             | $H L^{-1}$        | Characteristic gradient                         | Eq. (11)                             |  |
| K <sub>c</sub>                    | $H L^{-2}$        | Characteristic curvature                        | Eq. (13)                             |  |
| $A_c$                             | $L^2$             | Characteristic area                             | Eq. (15)                             |  |
| K' x' l' ata                      |                   | Parameters, variables, scales, etc., of the     | Eq. (18)                             |  |
| $K, x, t_c, ccc.$                 |                   | second of a pair rescaled landscapes            |                                      |  |
| $u_c$                             | $L T^{-1}$        | Characteristic horizontal velocity              | Eq. (20)                             |  |
| $h_I$                             | Н                 | Incision height                                 | Eq. (21)                             |  |
| $h_D$                             | Н                 | Diffusion height                                | Eq. (22)                             |  |
| $k_s$                             | Н                 | Steepness index                                 | $k_s = A^{m/n}  \nabla z $           |  |
| <i>K</i> <sub>thr</sub>           | $H L^{-2}$        | Curvature threshold for valley definition       | Sect. 4.1.4                          |  |
| h <sub>thr</sub>                  | Н                 | Incision height threshold for valley definition | $h_{thr} = l_c^2 \kappa_{thr} + h_c$ |  |
| Pe                                | [-]               | Péclet number                                   | Eq. (36)                             |  |
| С                                 | L T <sup>-1</sup> | Kinematic wave celerity                         | $c = K\sqrt{A}$                      |  |
| l                                 | L                 | Length scale                                    | Sect. 4.2                            |  |
| $t_I$                             | Т                 | Incision time                                   | Eq. (34)                             |  |
| t <sub>D</sub>                    | Т                 | Diffusion time                                  | Eq. (35)                             |  |
| р                                 | [-]               | Exponent of drainage area in scaling            | Sect. 4.2.3                          |  |
|                                   |                   | relationship with flow path length              |                                      |  |

**Table 2:** Parameters and resulting characteristic scales of the three landscapes presented in Figs. 1–5. Landscape B, the baseline landscape, has parameters with typical values (e.g., Perron et al., 2008; Tucker, 2009; Clubb et al., 2016). We use the baseline landscape to demonstrate properties of height and length scales in Figs. 7–10.

| Parameters,           | Units             | Landscape A      | Landscape B | Landscape C          |
|-----------------------|-------------------|------------------|-------------|----------------------|
| Characteristic scales |                   |                  | (Baseline)  |                      |
| K                     | a <sup>-1</sup>   | 10-5             | 10-6        | 10-7                 |
| Incision coefficient  |                   | -                | -           | -                    |
| D                     | $m^2 a^{-1}$      | 10 <sup>-3</sup> | 10-2        | 10 <sup>-1</sup>     |
| Diffusion coefficient | III a             | 10               | 10          | 10                   |
| U                     | m o <sup>-1</sup> | 10-5             | 10-4        | $2.5 \times 10^{-5}$ |
| Uplift rate           | III a             | 10               | 10          | 2.5x10               |

| $l_c = \sqrt{D/K}$<br>Characteristic length                  | m                 | 10              | 100             | 1,000                |
|--------------------------------------------------------------|-------------------|-----------------|-----------------|----------------------|
| $h_c = U/K$<br>Characteristic height                         | m                 | 1               | 100             | 250                  |
| $t_c = 1/K$<br>Characteristic time                           | а                 | 10 <sup>5</sup> | $10^{6}$        | 10 <sup>7</sup>      |
| $A_c = l_c^2$<br>Characteristic area                         | m <sup>2</sup>    | 10 <sup>2</sup> | 10 <sup>4</sup> | 10 <sup>6</sup>      |
| $G_c = h_c / l_c$<br>Characteristic gradient                 | m m <sup>-1</sup> | 0.1             | 1               | 0.25                 |
| $\kappa_c = S_c/l_c = h_c^2/l_c$<br>Characteristic curvature | m m <sup>-2</sup> | 10-2            | 10-2            | 2.5x10 <sup>-4</sup> |

# Figures