# Peer review of "Scaling and similarity of a stream-power incision and linear diffusion landscape evolution model"

_Earth Surface Dynamics, 2018_

## Referee Comment (RC1) · G.E. Tucker (Referee) · 14 May 2018

Overview:

This paper presents an insightful analysis of the simplest form of governing equation for drainage basin evolution. The authors present a dimensional analysis of the equation, showing that by treating horizonal and vertical scales separately, and by using the parameters themselves as the basis for normalization, one arrives at a simple and elegant dimensionless rendering—which contains important lessons about similarity. From the analysis emerge three characteristic scales, representing horizontal length, vertical distance (height), and time, respectively. And from this scale analysis emerge

valuable insights into a number of different aspects of basin and landscape evolution, such as: what sets the scale of transition between hillslopes and valleys? How do the efficiencies of hillslope transport and channel incision influence rates of drainage-divide migration? How are process parameters, which represent relative efficiencies of a geomorphic processes, manifested in quantitative terrain metrics? This is a theoretical exercise that shows very nicely how we can usefully think about the implications of the "stream power plus diffusion" model of landscape evolution: what are the key scales to consider, and how the parameters relate to one another. And although this ground has been covered before to some extent (most notably in the work of Garry Willgoose and colleagues from 1991), the present paper represents a valuable advance because it shows that one can capture the fundamental scaling of the system purely with reference to the process parameters.

I really appreciate that the authors took the time to craft an impeccable manuscript. The writing is lucid, the scope is thorough, and the arguments are well articulated. In fact, I've rarely seen such a flawless presentation in a first-time submission. Well done!

The decision to focus on the special case of m=1/2, n=1, while detailing the more general case in an appendix, strikes me as a sensible approach. The special case turns out to be beautifully simple. The general case is more complicated, and if they had tried to cover it in the main text the result would have been harder to penetrate. Yet they show that the fundamental insights derived from the simpler case usually hold for the more general one as well.

Specific comments keyed to text by page / line number:

1/8 Just an observation that we have a bit of a terminology problem as a community, which is illustrated here with the use of "Landscape Evolution Model" to refer to a set of governing equations. The word "model" is variously used to describe qualitative concepts, equation sets, numerical approximation methods and algorithms that represent certain processes, and particular computer programs that implement those numerics.

Of course, I don't expect the authors to solve this problem! But given that "LEM" has been used in the past to describe numerical models, you might consider specifying "widely used equation set for landscape evolution models (LEMs)..."

2/ 28 suggest "describes" rather than "simulates" (e.g., it would seem odd to say that mg "simulates" the force of gravity on an object at earth's surface, though if you calculated in a computer program that graphically depicts a falling rock, that might reasonably be called a simulation)

3/8 "in keeping with the stream power law": there are some devils in the details here; maybe simplest to summarize as "in keeping with the stream power law and the common assumption that discharge per unit width scales with the square root of drainage area"

3/17-18 this is just a minor terminology issue, but for what it's worth I think of soil creep as a general transport phenomenon that can be caused by the various processes listed, and that is often described quantitative using a diffusion model. To say "diffusive process" mixes the process and the model thereof. To list soil creep as a process comparable to bioturbation somehow mixes a possible causative agent (bioturbation) with the phenomenon that results (soil creep). Consider re-wording.

3/23-25 nice job being clear about baselevel lowering being mathematically equivalent to "uplift", which therefore needn't be caused by some kind of odd hillslope-scale vertical tectonics!

4/7-8 I think it would be fair to also cite Willgoose et al. 1991b (WRR, "part 2") here. I think of this work as the first serious introduction of dimensional analysis applied to landscape evolution, and he and his colleagues Ignacio Rodriguez-Iturbe and Rafael Bras deserve credit for leading the way.

eq 16 and following paragraph: beautiful!

6/21 this is an odd result, which seems to neglect the fact that dimensionless drainage

area can in principle range over orders of magnitude—thus, it's possible to have elements of the solution domain in which either the 1st or 2nd term on RHS dominates... which the next paragraph acknowledges.

8/12 why 34 in particular? Was this a systematic experimental design?

8/12-23 it would be helpful to state whether the model was actually run in dimensional form and then the output rescaled (which is probably the case). The sentence "We rescaled..." implies that dimensional grid coordinates were converted into dimensionless equivalents, but it is not explicitly stated whether this rescaling was applied to the inputs (i.e., input grid and K, D, U) or the outputs.

8/25-26 same question arises here: "run on rescaled versions of the same random TIN" is ambiguous because it seems to suggest that you took a dimensional TIN and rendered it dimensionless before conducting each run. I'm guessing that what was actually done was to calculate ahead of time, for each set of parameters, what horizontal and spatial dimensions would produce a size of 200x400 lc and 0-0.1 hc, and then set up dimensional model runs using these spatial dimensions. Is that correct? The alternative I suppose is that the inputs were re-scaled, meaning you used units of lc, hc, and tc instead of meters and years; but if that were the case, the initial grid would be always exactly the same in horizontal point locations, with the only differences being in the vertical scale.

9/1 I like how you varied lc and hc independently in these illustrations

10/19-28 wonderful insight into the divide migration problem—nice example of how much can be learned from dimensional analysis

section 4.1.1 - Here, interpretations for height scales are offered in terms of the characteristic time scale. It would be useful therefore to suggest an intuitive explanation for what tc represents. Given the definition of tc as 1/K, one such explanation is that it is the time required for one unit of incision given one unit area-slope quantity. "Unit area-slope

quantity", or "unit steepness", of course includes all cases of sqrt(A) S = 1 (eg, S=1 at A=1m2, S=0.1 at A=100m2, S=0.01 at A=10,000m2, S=0.001 at A=1,000,000m2). There's a nice rule of thumb in here: if I did the math right, unit steepness includes a slope of one-in-a-hundred at a drainage area of 1 hectare, or one-in-a-thousand at a drainage area of 1 km2.

eq 27: suggest pointing out to readers that this is just a rearrangement of the expected slope-area relationship for channels at steady state—thus, one recaptures this important relation.

fig 6: this is a nice illustration. If I were to change anything, it would simply be to plot -h_l (i.e., include the minus sign implied by eq 29 in relation to h_c and h_D).

15/10-17 This is a nice testable prediction. I can imagine issues arising related to the identification of drainage area (e.g., depending on the routing scheme used, at high res, many points within a valley could register as low A even though they are morphologically influenced by channelized flow), but worth trying.

22/16-17 suggest making these separate sentences

C1.2 why was it necessary to generate TINs using Matlab rather than just relying on CHILD's own TIN generation routines?

---

## Referee Comment (RC2) · Anonymous Referee #2 · 25 Jun 2018

This paper contains a clever and thorough elaboration on dimensional analysis of modeled hillslope-channel interactions. I enjoyed reading it, and I expect it will be influential. The key advance is the recasting of a governing equation for landscape evolution due to uniform rock uplift, linearly slope-dependent soil creep, and stream power-dependent channel incision in a dimensionless form with no parameters, which allows the rescaling of a single solution (for a given set of initial and boundary conditions) to any set of dimensional parameters. I have a few comments for the authors to consider as they revise their manuscript, but I recommend publication without re-review.

The authors might consider commenting on the limits of rescaling the linear diffusion

term to very steep slopes.

What about channel width? Presumably channels in this model are assumed to be "sub-grid-scale"; how is that taken into account in the governing equation or the numerical scheme?

Section 4.1.1: I suspect that this framework could quantify the independent controls on elevation contour shapes and vertical relief noted by Howard (1997), Tucker and Bras (1998), and Perron et al. (2008). Perhaps worth discussing.

P16 L28-29; Fig. 9: Absent any tectonic deformation that creates a shifting topographic divide, it is a geometric fact that a drainage divide migrates if and only if erosion rates differ across the divide, and that the sign of the difference in erosion rates determines the direction of drainage divide migration. I wouldn't characterize this as a research finding.

Section 4.2.2: The use of the flow path length (which depends on topography) as the horizontal length scale in the spatially variable Peclet number seems to run counter to the spirit of the non-dimensionalization that is the paper's centerpiece, in which the authors avoided using any length scales that are topographic outcomes of the model evolution. I understand the authors' reason for doing the calculation this way (Fig. 11), but it does make it difficult to use this definition of the Peclet number to predict model outcomes.

The summary and conclusions section is a bit long.

The inclusion of the dimensional analysis for the more general case of 2m != n is nice. Although it is understandably relegated to an appendix, this analysis considerably broadens the applicability of the paper.

---

## Editor Comment (EC1) · J. Braun (Editor) · 27 Jun 2018

I would like to congratulate the authors for a very interesting manuscript that I have really enjoyed reading. It provides a very insightful analysis of the main equation used to model landscape evolution. Both reviewers agree that the material covered should be published and propose comments and suggestions that should require minor revisions only. I therefore urge the authors to prepare a revised version of their manuscript. My recommendation is that it should be published in ESURF.

---

## Author Comment (AC1) · 20 Jul 2018

**Reply to two referees' comments on "Scaling and similarity of a stream-power incision and linear diffusion landscape evolution model" by Theodoratos et al.**

We are grateful to referee Greg Tucker and the anonymous referee for their feedback on our manuscript. In the following response to their reviews, we first quote their comments (in indented blocks of italic text) and then respond (in normal text). In the quoted comments of the referees, the numbering of pages, lines, and sections refers to the submitted manuscript, and in our responses, the numbering follows the revised manuscript with marked-up changes.

Nikos Theodoratos, Hansjörg Seybold, and James Kirchner

**1. Response to referee G.E. Tucker**

> *Overview:*
>
> *This paper presents an insightful analysis of the simplest form of governing equation for drainage basin evolution. The authors present a dimensional analysis of the equation, showing that by treating horizonal and vertical scales separately, and by using the parameters themselves as the basis for normalization, one arrives at a simple and elegant dimensionless rendering—which contains important lessons about similarity. From the analysis emerge three characteristic scales, representing horizontal length, vertical distance (height), and time, respectively. And from this scale analysis emerge valuable insights into a number of different aspects of basin and landscape evolution, such as: what sets the scale of transition between hillslopes and valleys? How do the efficiencies of hillslope transport and channel incision influence rates of drainage divide migration? How are process parameters, which represent relative efficiencies of a geomorphic processes, manifested in quantitative terrain metrics? This is a theoretical exercise that shows very nicely how we can usefully think about the implications of the "stream power plus diffusion" model of landscape evolution: what are the key scales to consider, and how the parameters relate to one another. And although this ground has been covered before to some extent (most notably in the work of Garry Willgoose and colleagues from 1991), the present paper represents a valuable advance because it shows that one can capture the fundamental scaling of the system purely with reference to the process parameters.*

Thank you for summarizing our manuscript in a way that nicely captures our intended message.

> *I really appreciate that the authors took the time to craft an impeccable manuscript. The writing is lucid, the scope is thorough, and the arguments are well articulated. In fact, I've rarely seen such a flawless presentation in a first-time submission. Well done!*

Thank you for acknowledging the effort that we invested!

> *The decision to focus on the special case of m=1/2, n=1, while detailing the more general case in an appendix, strikes me as a sensible approach. The special case turns out to be beautifully simple. The general case is more complicated, and if they had tried to cover it in the main text the result would have been harder to penetrate. Yet they show that the*

*fundamental insights derived from the simpler case usually hold for the more general one as well.*

Thank you for supporting this approach. Presenting the general case in an appendix was an efficient compromise between the two alternative options of presenting it in the main text of this manuscript or in a separate manuscript. The former option would have made the manuscript very hard to read, while the latter could have led readers to reasonably doubt the general validity of our results. An important, additional objective of including the general case in this manuscript was to help readers avoid conclusions that are valid only for the special case of $m = 0.5$ and $n = 1$. For instance, the characteristic height, defined as $h_c = U/K$ (Eq. 4) for the special case, should not be interpreted as expressing the competition between only uplift and incision, because in the general case it also depends on the diffusion coefficient $D$ (Eq. A9). See further below a similar discussion regarding the characteristic time $t_c$.

> *Specific comments keyed to text by page / line number:*
>
> *1/8 Just an observation that we have a bit of a terminology problem as a community, which is illustrated here with the use of "Landscape Evolution Model" to refer to a set of governing equations. The word "model" is variously used to describe qualitative concepts, equation sets, numerical approximation methods and algorithms that represent certain processes, and particular computer programs that implement those numerics. Of course, I don't expect the authors to solve this problem! But given that "LEM" has been used in the past to describe numerical models, you might consider specifying "widely used equation set for landscape evolution models (LEMs)..."*

We agree, and we changed our wording in page 1, line 8, and page 2, line 14.

> *2/ 28 suggest "describes" rather than "simulates" (e.g., it would seem odd to say that mg "simulates" the force of gravity on an object at earth's surface, though if you calculated in a computer program that graphically depicts a falling rock, that might reasonably be called a simulation)*

We followed this suggestion in the revised manuscript (page 2, line 32).

> *3/8 "in keeping with the stream power law": there are some devils in the details here; maybe simplest to summarize as "in keeping with the stream power law and the common assumption that discharge per unit width scales with the square root of drainage area"*

We followed your recommendation in the revised manuscript (page 3, lines 11–12).

> *3/17-18 this is just a minor terminology issue, but for what it's worth I think of soil creep as a general transport phenomenon that can be caused by the various processes listed, and that is often described quantitative using a diffusion model. To say "diffusive process" mixes the process and the model thereof. To list soil creep as a process comparable to bioturbation somehow mixes a possible causative agent (bioturbation) with the phenomenon that results (soil creep). Consider re-wording.*

We have re-worded this sentence in a way that we hope is clearer (page 3, line 23).

> *3/23-25 nice job being clear about baselevel lowering being mathematically equivalent to "uplift", which therefore needn't be caused by some kind of odd hillslope-scale vertical tectonics!*

Thank you.

> *4/7-8 I think it would be fair to also cite Willgoose et al. 1991b (WRR, "part 2") here. I think of this work as the first serious introduction of dimensional analysis applied to landscape evolution, and he and his colleagues Ignacio Rodriguez-Iturbe and Rafael Bras deserve credit for leading the way.*

Our statement in these lines does not refer in general to dimensional analyses of LEMs. Rather, it refers specifically to dimensional analyses that explore the scales of regime transitions in LEMs that do not distinguish a priori between these regimes. The model by Willgoose et al. (1991) does distinguish between regimes and, thus, we did not cite it as an example of these specific dimensional analysis. We revised the manuscript (page 4, lines 16–17) so that it is clearer that we refer to specific cases of dimensional analyses.

However, we too view G. Willgoose and his colleagues as having led dimensional analysis of LEMs and, based on your comment, we realize that our view could be communicated more clearly. Thus, we added an additional citation of Willgoose et al. (1991) in page 2, line 20 of the revised manuscript, where we clarify what is and what is not new in our study.

> *eq 16 and following paragraph: beautiful!*

Thank you.

> *6/21 this is an odd result, which seems to neglect the fact that dimensionless drainage area can in principle range over orders of magnitude—thus, it's possible to have elements of the solution domain in which either the 1st or 2nd term on RHS dominates... which the next paragraph acknowledges.*

Based on your comment, we realized that our statement "none of the terms of this LEM (Eq. 1) can be neglected" should have included the qualifier "everywhere". Thus, we changed our statement to "none of the terms of this LEM (Eq. 1) is negligible everywhere across a landscape" and we added two sentences that further clarify this statement (page 7, lines 21–24).

> *8/12 why 34 in particular? Was this a systematic experimental design?*

No, 34 just happens to be the number of simulations that we ran. We mention this number to clarify that we did not run only the three simulations that we present in Figs. 1–5. In the revised manuscript, in page 9, line 12, we wrote "multiple combinations" instead of "34 combinations" so readers will not wonder whether this number is important. Further below, in page 9, line 15, we mention that the combinations were 34, but in a way that we hope will not give the impression that this specific number is important.

> *8/12-23 it would be helpful to state whether the model was actually run in dimensional form and then the output rescaled (which is probably the case). The sentence "We rescaled..." implies that dimensional grid coordinates were converted into dimensionless equivalents, but it is not explicitly stated whether this rescaling was applied to the inputs (i.e., input grid and K, D, U) or the outputs.*

See our response to your next comment.

> *8/25-26 same question arises here: "run on rescaled versions of the same random TIN" is ambiguous because it seems to suggest that you took a dimensional TIN and rendered it*

*dimensionless before conducting each run. I'm guessing that what was actually done was to calculate ahead of time, for each set of parameters, what horizontal and spatial dimensions would produce a size of 200x400 lc and 0-0.1 hc, and then set up dimensional model runs using these spatial dimensions. Is that correct? The alternative I suppose is that the inputs were re-scaled, meaning you used units of lc, hc, and tc instead of meters and years; but if that were the case, the initial grid would be always exactly the same in horizontal point locations, with the only differences being in the vertical scale.*

Yes, it is correct that we calculated ahead of time what horizontal and vertical lengths would lead to the desired domain sizes. We realized that we did not sufficiently describe how we set up the rescaled TINs, and we thank you for this question. We added a description of the procedure that we followed in Appendix C1.2 of the revised manuscript (page 30, lines 13–29) and a pointer to that description in the main text (page 9, lines 17–18). Below, we describe the workflow that we developed in some more detail than in the revised manuscript, because we assume that you would be interested to read how we used your CHILD model to implement our modeling approach.

1. Using Matlab, we generated random TINs with $x$-coordinates between 0 and 200, $y$-coordinates between 0 and 400, average mesh edge of 0.8, and $z$-coordinates drawn from a uniform distribution between 0 and 0.1. Thus far, these values are just numbers without any units. In other words, these TINs are dimensionless. We generated these TINs using Matlab to have control of the next steps of our procedure.

2. For each combination of the parameters $K$, $D$, and $U$, we calculated the characteristic length and height scales $l_c$ and $h_c$ (Eqs. 3 and 4). In parallel, for each parameter combination, we calculated the CHILD input parameter KB according to Eq. (C2) (denoted as $k_b$ in that equation) and used $D$ and $U$ as the CHILD input parameters KD and UPRATE.

3. Using Matlab, we multiplied the $x$- and $y$-, and $z$-coordinates of the dimensionless TINs by the calculated characteristic length and height scales $l_c$ and $h_c$, respectively. This yielded the dimensional TINs corresponding to each parameter combination. CHILD assumes that $x$, $y$, and $z$ are in units of meters, therefore, we calculated $l_c$ and $h_c$ in units of meter as well. We exported the dimensional TINs as .txt files to be used as inputs by CHILD.

4. For each parameter combination, we created a .in CHILD input file that included the calculated KB, KD, and UPRATE parameters, and a pointer to the .txt file with the corresponding dimensional TIN (entered as parameter POINTFILENAME).

Note that we automated the above workflow to ensure that each parameter combination would reliably use the correct dimensional TINs. Thus, for each combination of the parameters $K$, $D$, and $U$, the values of the CHILD parameters KB, KD, UPRATE, and POINTFILENAME were automatically written into a .in CHILD input file that was automatically named according to that combination. Likewise, for each combination of $K$, $D$, and $U$, the .txt TIN file that was exported by Matlab was automatically named. Finally, the .in and .txt files corresponding to each parameter combination were automatically placed together in an automatically-named subfolder, where a call to the compiled *child* program and the corresponding .in input file was automatically issued.

> *9/1 I like how you varied lc and hc independently in these illustrations*

Thank you.

*10/19-28 wonderful insight into the divide migration problem—nice example of how much can be learned from dimensional analysis*

Thank you.

*section 4.1.1 - Here, interpretations for height scales are offered in terms of the characteristic time scale. It would be useful therefore to suggest an intuitive explanation for what tc represents. Given the definition of tc as 1/K, one such explanation is that it is the time required for one unit of incision given one unit area-slope quantity. "Unit area-slope quantity", or "unit steepness", of course includes all cases of sqrt(A) S = 1 (eg, S=1 at A=1m2, S=0.1 at A=100m2, S=0.01 at A=10,000m2, S=0.001 at A=1,000,000m2). There's a nice rule of thumb in here: if I did the math right, unit steepness includes a slope of one-in-a-hundred at a drainage area of 1 hectare, or one-in-a-thousand at a drainage area of 1 km2.*

You are correct that the characteristic time $t_c$ "is the time required for one unit of incision given one unit area-slope quantity." However, we would like to add some precautionary comments.

First, $t_c$ depends only on the incision coefficient $K$ (Eq. 5) only in the case of $m = 0.5$ and $n = 1$; in general, $t_c$ depends on all three parameters $K$, $D$, and $U$ (Eq. A10). Therefore, we think it would be misleading to pin an explanation of $t_c$ exclusively on the incision term. Indeed, $t_c$ is also the time required for one unit of deposition (or erosion) given one (or minus one) unit of curvature, and for one unit of elevation increase given one unit of uplift rate.

Furthermore, explaining $t_c$ in terms of unit elevation change could lead to circular logic. Specifically, taking uplift as an example, the characteristic height $h_c$ is the elevation uplifted during one unit of characteristic time $t_c$, which is the time needed to uplift one unit of $h_c$, which is… Therefore, we can only explain $h_c$ in terms of $t_c$, or $t_c$ in terms of $h_c$, but not both. We chose the former for reasons that we explain in the following paragraphs.

To summarize Sect. 4.1, we multiplied by $t_c$ the rates of elevation change due to incision, diffusion, and uplift, i.e., the three RHS terms of Eq. (1), to express them in units of height. We used the resulting height scales to derive topographic relations that express competitions between processes. If we had used some other, arbitrary time interval $\Delta t$ instead of $t_c$, we would have obtained the exact same topographic relations, because the multiplications that convert rates of elevation change into heights are all linear in time. For instance, the steady-state incision–uplift balance, i.e., the condition $K\sqrt{A}|\nabla z|\,\Delta t = U\,\Delta t$, is satisfied at points with $\sqrt{A}|\nabla z| = U/K$ and $\nabla^2 z = 0$ for any $\Delta t$.

In contrast, in the case of the horizontal direction, which we presented in Sect. 4.2, the incision time $t_I$ scales linearly with the length scale $l$ (Eq. 34), but the diffusion time $t_D$ scales with $l^2$ (Eq. 35). Therefore, the length scale $l$ is not eliminated from the definition of the Péclet number (Eq. 36). As it turns out, an implication of this is that at points with $l \approx l_c$ and $A \approx A_c$, where $\mathrm{Pe} \approx 1$, the incision and diffusion time scales are such that $t_I \approx t_D \approx t_c$ (page 19, line 21–23 of the revised manuscript). In other words, the horizontal advection–diffusion balance leads to conditions that give explanations to both $l_c$ and $t_c$.

Consequently, focusing on the vertical direction (Sect. 4.1), we interpreted the characteristic height $h_c$ in terms of the characteristic time $t_c$, which we interpreted – along with the characteristic length $l_c$ – by focusing on the horizontal direction (Sect. 4.2). In the opening of Sect. 4.2 of the revised manuscript, we added a sentence stating that we can use the Péclet number to interpret $l_c$ and $t_c$ (page 18, lines 17–19).

*eq 27: suggest pointing out to readers that this is just a rearrangement of the expected slope-area relationship for channels at steady state—thus, one recaptures this important relation.*

We compare Eq. (27) to the slope–area relationship for steady-state channels in Sect. 4.1.3 (page 17, lines 1–8). We added a pointer to this comparison at the end of the paragraph that introduces Eq. (27) (page 15, lines 2–3).

*fig 6: this is a nice illustration. If I were to change anything, it would simply be to plot -h_I (i.e., include the minus sign implied by eq 29 in relation to h_c and h_D).*

Thank you for the suggestion. However, we are plotting $+h_I$ to illustrate that the $h_I$ and $h_D$ curves have a constant distance that is equal to $h_c$, i.e., to graphically illustrate Eq. (29).

*15/10-17 This is a nice testable prediction. I can imagine issues arising related to the identification of drainage area (e.g., depending on the routing scheme used, at high res, many points within a valley could register as low A even though they are morphologically influenced by channelized flow), but worth trying.*

Thank you. We agree that this would be an interesting research question, but for future work.

*22/16-17 suggest making these separate sentences*

Thank you for the suggestion. We separated these sentences in the revised manuscript (page 23, lines 30–31).

*C1.2 why was it necessary to generate TINs using Matlab rather than just relying on CHILD's own TIN generation routines?*

We mainly used Matlab to control and automate the preparation of input files for CHILD as described above.

**2. Response to the anonymous referee**

*This paper contains a clever and thorough elaboration on dimensional analysis of modeled hillslope-channel interactions. I enjoyed reading it, and I expect it will be influential. The key advance is the recasting of a governing equation for landscape evolution due to uniform rock uplift, linearly slope-dependent soil creep, and stream power-dependent channel incision in a dimensionless form with no parameters, which allows the rescaling of a single solution (for a given set of initial and boundary conditions) to any set of dimensional parameters. I have a few comments for the authors to consider as they revise their manuscript, but I recommend publication without re-review.*

Thank you.

*The authors might consider commenting on the limits of rescaling the linear diffusion term to very steep slopes.*

The linear diffusion term of Eqs. (1) and (16) cannot describe real landscapes with very steep hillslopes, even though mathematically it can take any non-negative real number as the slope. We

added a comment about this in Sect. 2.1 after the discussion of the types of landscapes which Eq. (1) is presumed to describe (page 3, line 39, to page 4, line 2).

> *What about channel width? Presumably channels in this model are assumed to be "sub-grid-scale"; how is that taken into account in the governing equation or the numerical scheme?*

We thank you for this question, because it connects our study with an important discussion that has been ongoing in the literature concerning this family of LEMs.

Specifically, Pelletier (2010) discusses the resolution-dependence of flow-routing algorithms, and thus of LEMs that use these algorithms. To minimize this dependence, Pelletier (2010) recommends including the factor $\delta/w$ in the diffusion term, where $\delta$ is the grid resolution and $w$ is the flow width. Similarly, Perron et al. (2008) included the inverse factor, i.e., $w/\delta$, in the incision term.

Our CHILD simulations did not include such factors. However, our rescaling approach guaranteed that the resolution-dependence was consistent and reproducible across rescaled simulations. Specifically, across rescaled landscapes, both $\delta$ and $w$ scaled with $l_c$. Therefore, corresponding points of rescaled landscapes had equal $\delta/w$ ratios, and maintained consistent dependence on resolution. In the case of landscapes that are not rescaled, however, the dependence on resolution would not be consistent.

(Note that in the case of CHILD, which uses TINs, $\delta$ would correspond to some characteristic TIN widths, e.g., to lengths of edges of Voronoi polygons. Furthermore, note that, although Eq. (1) does not explicitly include flow widths, we can implicitly assume how flow widths would scale with drainage area, because the derivation of Eq. (1) is based on assumed relationships between flow width and drainage area; e.g., Tucker et al., 2001; Perron et al., 2008.)

In the revised manuscript (page 7, lines 4–18), we addressed your question by adding a discussion about various factors that control the solution of Eq. (1) in ways that are not apparent just by looking at Eq. (16), the dimensionless form of Eq. (1). Specifically, we pointed out that alternative dimensionless forms of Eq. (1) are able to reveal some properties of the LEM that are not revealed by Eq. (16). One such example is the resolution-dependence, as described above. Other examples are the dependence of solutions on boundary conditions, initial conditions, or on the size of the domain. For instance, Eq. (19) of Perron et al. (2008) includes the size of the domain ($l$ in their notation) and the steady-state total relief ($\zeta$); therefore, it explicitly shows that solutions depend on the size of the domain and the initial conditions (given that, for a given landscape, $\zeta$ depends on the initial conditions).

> *Section 4.1.1: I suspect that this framework could quantify the independent controls on elevation contour shapes and vertical relief noted by Howard (1997), Tucker and Bras (1998), and Perron et al. (2008). Perhaps worth discussing.*

Indeed, one could use equations of Sect. 4.1 (e.g., Eqs. (29), (30), or (31)) to show that, in the case of Eq. (1), if the characteristic height $h_c$ is changed, while the characteristic length $l_c$ is kept constant, then a landscape can be rescaled only vertically and not horizontally, i.e., the vertical components of the variables will be rescaled, while the horizontal components will remain constant. Note, however, that in the case of the general LEM (Eq. A1), horizontal and vertical components cannot in general be rescaled independently. The definitions of characteristic scales of the general LEM (Eqs. A8–A10) reveal the values of the exponents $m$ and $n$ for which horizontal, vertical, or temporal components of variables can be controlled independently, as shown, for example, in the discussion in page 25, lines 6–15.

*P16 L28-29; Fig. 9:* *Absent any tectonic deformation that creates a shifting topographic*
*divide, it is a geometric fact that a drainage divide migrates if and only if erosion rates*
*differ across the divide, and that the sign of the difference in erosion rates determines the*
*direction of drainage divide migration. I wouldn't characterize this as a research finding.*

The mechanisms controlling divide migration have been the subject of considerable discussion in the recent geomorphological literature. While exploring the properties of the incision height $h_I$, which is equal to the steepness index $k_s$ in the case of Eq. (1), we happened to reproduce the result of Whipple et al. (2017), that the steepness index (among several other metrics) predicts the direction of ridge migration (it is Whipple et al. who characterize this as a result, not us). We believe this is worth reporting, without wading into the argument between those who use the steepness index and those who use $\chi$ and other metrics.

As you wrote, the direction of divide migration depends on the difference of erosion rates across the divide. Near the ridge of a landscape that follows Eq. (1), erosion is caused by both the incision and the diffusion term. Thus, we found it interesting that, in our simulations, the direction of divide migration could be predicted using the incision height $h_I$ on its own, i.e., without taking into account the diffusion height $h_D$. This is in line with Willet et al.'s (2014) assumption that the reorganization of drainage basins could be studied by focusing only on river channels, leaving hillslopes out of the analysis. However, this interesting observation is beyond the scope of the present study and, thus, we did not include it in our manuscript, because we did not want to assert something that we have not proved.

> *Section 4.2.2: The use of the flow path length (which depends on topography) as the*
> *horizontal length scale in the spatially variable Peclet number seems to run counter to*
> *the spirit of the non-dimensionalization that is the paper's centerpiece, in which the*
> *authors avoided using any length scales that are topographic outcomes of the model*
> *evolution. I understand the authors' reason for doing the calculation this way (Fig. 11),*
> *but it does make it difficult to use this definition of the Peclet number to predict model*
> *outcomes.*

(Note that the subsection that follows Sect. 4.2.2 was mistakenly numbered as 4.2.4 in the submitted manuscript. In the revised manuscript, we corrected this mistake, and in this response, we refer to this section as 4.2.3.

We thank you for this comment, because it gives us an opportunity to clarify the motivation behind our key assumptions and methodologies.

Using the flow path length to calculate values of the Péclet number does not run counter to the spirit of our analysis. We avoided using topography-dependent scales to define the characteristic scales. To interpret the characteristic scales, however, we introduced process-specific scales that vary across the landscape, such as the incision and diffusion heights $h_I$ and $h_D$ (Eqs. 21 and 22). Then, we explored whether there is anything special about the points at which the process-specific scales become equal to the characteristic scales or to each other.

As we explain further above in our response to referee Greg Tucker, we found that we could interpret either the characteristic height $h_c$ in terms of the characteristic time $t_c$, or $t_c$ in terms of $h_c$, but not both. We found, however, that we could interpret $t_c$ together with the characteristic length $l_c$ by including both the drainage area $A$ and a length scale $l$ in the definition of the incision time $t_I$ (Eq. 34). This revealed that advective and diffusive propagation of perturbations lead to different scalings between

time and length scales. Therefore, we could interpret $t_c$ and $l_c$ in terms of the competition between advective and diffusive propagation of perturbations, and, in turn, interpret $h_c$ in terms of $t_c$.

Thus far, $l$ could be any arbitrary length scale. However, to calculate values of the Péclet number across a landscape, a specific length scale should be chosen. In Sect. 4.2.2 we chose flow path length to calculate Péclet number values for Fig. 10, a figure which suggests that defining the Péclet number according to Eq. (36) is reasonable, while in Sect. 4.2.3 we discussed the similarities and differences between our Péclet number definition and that of Perron et al. (2008, 2009, 2012). The Péclet number, like the incision time $t_I$, cannot be (usefully) defined in terms of the model parameters alone.

Generally, we think that there is no single Péclet number definition that fits all problems. Rather, employing different definitions that use different length scales, we can study different properties of landscapes. This is analogous to studying turbulence by using different characteristic system sizes for different kinds of flows, e.g., flow in pipes vs. flow past a cylinder.

In the case of landscapes, one could study advective and diffusive propagation of perturbations using the flow path length, as discussed in our manuscript, but one could also characterize sizes of whole landscapes – relative to their parameters $K$ and $D$ – using the total size of each landscape as the length scale and defining a landscape-wide Péclet number. Small values of such a Péclet number would correspond to landscapes that consist mainly of hillslopes, because they are too small relative to the spacing of valleys that would be expected for the parameters of these landscapes. In contrast, large values of such a Péclet number would correspond to dissected landscapes, which are large enough to fit their valleys. The transition between these two cases need not occur at a Péclet number value of roughly one; this value would need to be discovered using numerical experiments (e.g., Perron et al., 2012).

To summarize our response to your comment, a key motivation behind Sect. 4.2 is to interpret the characteristic time $t_c$ independently of the characteristic height $h_c$. For this, we defined an incision time $t_I$ whose scaling with the length scale $l$ is different than that of the diffusion time $t_D$ (Eqs. 34 and 35). To illustrate that these two time scales define a reasonable Péclet number, we used the flow path length, which is a reasonable scale because it is in line with the way that perturbations are propagated by the incision and diffusion terms. Given that we modified Perron et al.'s (2008, 2009, 2012) original definition of the Péclet number, we presented a brief comparison of the two definitions, but a more thorough comparison would be beyond the scope of this study.

As we wrote further above in our response to referee Greg Tucker, we added a sentence stating that we can use the Péclet number to interpret $l_c$ and $t_c$ in page 18, lines 17–19.

> *The summary and conclusions section is a bit long.*

We agree that it is long, but we think that this is a feature rather than a bug: a manuscript with such length, complexity, and number of results, requires a long summary to be sufficiently summarized.

> *The inclusion of the dimensional analysis for the more general case of 2m != n is nice. Although it is understandably relegated to an appendix, this analysis considerably broadens the applicability of the paper.*

Thank you. See also our response to referee Greg Tucker regarding this appendix. As we explain there, relegating this analysis in the appendix was a compromise, which came with certain trade-offs. For instance, some conclusions of this analysis will inevitably receive less attention than they might otherwise receive.

**References**

[revised manuscript text omitted]

At drainage divides ($A = 0$), Eq. (A23) becomes

$$h_D = -h_c \quad \Leftrightarrow \quad D\nabla^2 z\, t_c = -U\, t_c \quad \Leftrightarrow \quad \nabla^2 z = -U/D = -\kappa_c \,. \tag{A25}$$

Substituting the definition of $h_D$ (Eq. A20) into Eq. (A23) yields:

$$h_I = l_c^2 \nabla^2 z + h_c .$$
(A26)

i.e., $h_I$ remains a linear function of curvature in steady state. Thus, $h_I$ can be used for valley definition for the generic model in the same way that it can be used for the simplified model.

**A5 Length and time scales**

The celerity of the incision term is:
$$c(A, |\nabla z|) = K A^m (|\nabla z|)^{n-1} .$$
(A27)

[revised manuscript text omitted]

| | | | | |
|---|---|---|---|---|
| $t_c = 1/K$
Characteristic time | a | $10^5$ | $10^6$ | $10^7$ |
| $A_c = l_c^2$
Characteristic area | m$^2$ | $10^2$ | $10^4$ | $10^6$ |
| $G_c = h_c/l_c$
Characteristic gradient | m m$^{-1}$ | 0.1 | 1 | 0.25 |
| $\kappa_c = S_c/l_c = h_c^2/l_c$
Characteristic curvature | m m$^{-2}$ | $10^{-2}$ | $10^{-2}$ | $2.5 \times 10^{-4}$ |

**Figures**

**Figure 1: Horizontal similarity of rescaled landscapes.** Steady-state shaded-relief maps demonstrate the horizontal geometric similarity of three landscapes with widely varying parameters but properly rescaled domains (see Eq. (18) for definition of proper rescaling). We label axes in units of each landscape's characteristic length $l_c$ (on the bottom and right in normal fonts) and in kilometers (on the top and left in bold fonts) to highlight the similarity of the three landscapes despite their very different sizes. The landscapes are shown in order of increasing $l_c$ from left to right. Notice that the sizes of the three simulated domains differ by factors of 10 in units of kilometers, but are identical in units of $l_c$. Lengths and heights scale separately. This leads to different characteristic gradients $G_c$ across landscapes, manifested as varying grayscale intensity ranges. Despite these pronounced differences in gradients, the three landscapes are geometrically similar in planview.

**Figure 2: Vertical similarity of rescaled landscapes.** Steady-state elevation maps and transects demonstrate the vertical geometric similarity of the three landscapes of Fig. 1. We color the maps by elevation with color scales that are rescaled by each landscape's characteristic height $h_c$ using a colormap distributed with the SIGNUM model (Refice et al., 2012). Notice that across color scales each color corresponds to the same elevation value in units of $h_c$, but to different elevation values in meters. Therefore, the fact that the color patterns are identical reveals the vertical similarity of the landscapes. Transects corroborate the vertical similarity. Transects pass through the highest peak of the landscapes and are marked on the maps with thick black lines. Note that elevations measured in units of $h_c$ (in normal fonts) are the same, while elevations measured in meters (in bold fonts) are different. The landscapes are shown in order of increasing $l_c$ from left to right. (The ranges of elevation in map color scales and transect $z$ axes match those of Figs. 4 and 5.)

**Figure 3: Temporal similarity of evolving, rescaled landscapes.** We compare the evolution of the three landscapes of Fig. 1 using shaded-relief maps drawn at four properly rescaled moments in time (see Eq. (18 d) for definition of proper rescaling of time). The comparison shows that, at rescaled moments in time, the horizontal patterns of the landscapes are geometrically similar (and geometrically identical in units of $l_c$). Each row shows four snapshots of a given landscape. The fourth column shows steady-state landscapes, i.e., those of Fig. 1. The snapshots that appear in each vertical column correspond to the same moment in rescaled time. Values of time in units of $t_c$ are the same along each column (labels in normal fonts), while the values of time in years vary (labels in bold fonts). Time increases from left to right, and horizontal scale increases from top to bottom. Lengths and heights scale separately; this leads to different characteristic gradients $G_c$ across landscapes, manifested as varying grayscale intensity ranges across rows.

**Figure 4: Temporal, horizontal, and vertical similarity of evolving, rescaled landscapes.** We compare elevation maps of the three evolving landscapes of Fig. 2, using the same snapshots as in Fig. 3 and the same layout (i.e., landscapes sorted by row, rescaled times sorted by column). Color maps are rescaled by each landscape's characteristic height, $h_c$. The comparison illustrates the horizontal and vertical geometric similarity of the landscapes at rescaled moments in time. For each landscape (i.e., across each row) we use a single color scale, constant in time, to show how elevations (rescaled by each landscape's characteristic height $h_c$) evolve. The fact that the color patterns are identical within each column reveals the vertical component of the temporal and geometric similarity of the landscapes. Thick black lines mark the transects shown in Fig. 5. The fourth column shows steady-state landscapes, i.e., those of Fig. 2.

**Figure 5: Temporal and vertical similarity of evolving, rescaled landscapes.** Transects of the three evolving landscapes of Fig. 2 corroborate the vertical component of their temporal and geometric similarity. We use the same snapshots as in Figs. 3 and 4 with the same layout (i.e., landscapes sorted by row, rescaled times sorted by column). The rescaled transects are identical along each column, demonstrating the exact temporal and geometric similarity of the rescaled landscapes. Transects pass through the highest peak of the steady-state landscapes and are marked on the maps of Fig. 4 with thick black lines. The fourth column shows steady-state transects, i.e., those of Fig. 2.

**Figure 6: Schematic illustration of height scales.** Profiling the incision, diffusion, and characteristic height scales $h_I$, $h_D$, and $h_c$ along a flow path allows visualizing their properties as described by Eqs. (27)–(29). Part (a) shows the height scales of incision, diffusion, and uplift ($h_I$, $h_D$, and $h_c$, corresponding to the solid, dashed, and dotted black lines, respectively) along a flow path from the drainage divide to a valley. Part (b) shows these height scales as changes in elevation along a steady-state elevation profile (thick green line). Subtracting $h_I$ from the elevations along the profile, or adding $h_D$ or $h_c$ to them, shows the change in elevation per unit of characteristic time $t_c$ that would result from incision, diffusion, and uplift (the solid, dashed, and dotted gray lines, respectively). At the divide (point $P_1$), incision is ineffective and diffusion balances uplift ($h_I = 0$ and $h_D = -h_c$; Eq. 28). At the point where curvature is zero (point $P_2$), net diffusion is zero and incision balances uplift ($h_D = 0$ and $h_I = h_c$; Eq. 27). (Note that $P_2$, where $\nabla^2 z = 0$, generally is not the same as the inflection point of the profile line.) Along the entire profile, the combination of incision and diffusion balances uplift; thus, the distance between the $h_I$ and $h_D$ lines in part (a) (solid and dashed black lines) is constant and equal to $h_c$ ($h_I - h_D = h_c$; Eq. 29).

**Figure 7: Steady-state relationship between drainage areas, slopes, and curvatures, parameterized by characteristic length and height scales $l_c$ and $h_c$.** Plotting incision height data ($h_I = \sqrt{A}|\nabla z|$) versus curvature data ($\nabla^2 z$) from the simulated steady-state landscape B (shown in Figs. 1–5), shows that these data followed a linear trend consistent with the relation $\sqrt{A}|\nabla z| = l_c^2 \, \nabla^2 z + h_c$ (Eq. 31). We label incision height and curvature axes in units of m and m m$^{-2}$, respectively, in bold fonts, and in units of characteristic height $h_c$ and characteristic curvature $\kappa_c$, respectively, in normal fonts. Vertical and horizontal dashed lines facilitate the calculation of the slope and intercept of this trend. For every unit of $\kappa_c$ of curvature increase, the product $\sqrt{A}|\nabla z|$ increases by a unit of $h_c$; thus, the trend's slope is $h_c/\kappa_c = l_c^2$. To the curvature value $\nabla^2 z = 0$ corresponds the value $\sqrt{A}|\nabla z| = h_c$; thus, the trend's intercept is $h_c$. The linear relationship illustrated here shows that we can view the characteristic length and height $l_c$ and $h_c$ as parameters of a steady-state relationship that constrains the topographic properties $A$, $|\nabla z|$, and $\nabla^2 z$. This relationship must be satisfied so that incision, diffusion, and uplift are in equilibrium. Thus, we can view $h_c$ as a scale that expresses the competition between these three processes. (Note the two arrows that mark where data from points P1 and P2 of Fig. 6 would plot.)

**Figure 8: Steady-state valley networks visualized by incision height $h_I$.** The simulated equilibrium landscape B (shown in Figs. 1–5) is shown here with each pixel colored by the incision height $h_I = \sqrt{A}|\nabla z|$. In steady state, this incision height is linearly related to topographic curvature (Eq. (33), Fig. 7), with values of $h_I < h_c$ corresponding to convex topography (ridgelines) and values of $h_I > h_c$ corresponding to concave topography (valleys). Thus, high values of $h_I$ reveal the dendritic valley network.

**Figure 9: Prediction of ridge migration by differences of incision height $h_I$ across ridges.** These maps show that migrating ridges move toward the side with relatively smaller $h_I$ values. The top row shows four snapshots during the early phase of the evolution of landscape B (also shown in Figs. 1–5). The red squares seen in the top row are focused on a ridge that migrates from left to right; they are also shown magnified in the bottom row. The dashed black lines inside the red squares are fixed at the initial position of the migrating ridge to more clearly illustrate its movement. The black arrows in the bottom row point to the direction toward which the ridge will migrate. In all maps, each pixel is colored by the incision height $h_I = \sqrt{A}|\nabla z|$. Darker colors correspond to higher values of $h_I$. These maps show that drainage basins expanded at the expense of neighboring drainage basins with lighter colors, i.e., with relatively lower $h_I$ values. For the case of Eq. (1), the incision height $h_I$ is equal with the steepness index $k_s$, which was among the metrics used by Whipple et al. (2017) to quantify erosion rates. Therefore, these maps confirm Whipple et al.'s finding that the direction of ridge migration can be predicted by erosion rate differences across migrating ridges.

**Figure 10: Steady-state valley networks visualized by Péclet number Pe.** The simulated equilibrium landscape B (shown in Figs. 1–5) is shown here with each pixel colored by the logarithm of $Pe = \sqrt{A}\, l/l_c^2$ (Eq. 36), where $l$ is the flow path length (the maximum distance along flow paths from each pixel to the drainage divide). The Péclet number quantifies the relative horizontal influence of incision versus diffusion. Thus, dark pixels in this map are horizontally dominated by incision (i.e., valleys) and light pixels by diffusion (i.e., ridgelines).

**Figure 11: Dependence of drainage area on flow path length and convergence or divergence of topography.** Three points that have equal flow path lengths can have very different drainage areas. Elevation contours (gray lines) reveal that point $P_1$ is in a valley with convergent topography, $P_2$ on a hillslope with planar topography, and $P_3$ on an interfluve with divergent topography. The thick dashed line shows the ridge line and the three dotted lines show flow paths from the ridge to the points $P_1$, $P_2$, and $P_3$. The three flow paths have equal lengths $l$. The three brown polygons show the contributing areas that correspond to a given contour width (thick black bars) centered at the three points. Point $P_1$ has a much larger drainage area $A$ than points $P_2$ and $P_3$. Note the topography of the contributing area of point $P_1$; it tends to become planar near the ridge. Specifically, the lateral boundaries of the contributing area tend to become parallel, and the curvature of the contour lines decreases. This example highlights that, in general, contributing areas have mixed topographies.